



# Assessment of streamflow decrease due to climate vs. human influence in a semiarid area

Hamideh Kazemi[1,2], Hossein Hashemi[2], Fatemeh Fadia Maghsood[2], Seyyed Hasan Hosseini[2,3], Ranjan Sarukkalige[1], Sadegh Jamali[4], and Ronny Berndtsson[2]

[1]School of Civil and Mechanical Engineering, Curtin University, Perth, Australia
[2] Department of Water Resources Engineering & Center for Middle Eastern Studies, Lund University, Lund, Sweden
[3] Department of Water Engineering, Faculty of Agriculture, University of Tabriz, Tabriz, Iran
[4] Department of Technology and Society, Lund University, Lund, Sweden

*Correspondence to*: Hamideh Kazemi (hamideh.kazemi@postgrad.curtin.edu.au)
https://orcid.org/0000-0002-9088-0958.

**Abstract.** This paper uses the Budyko method to investigate mean annual streamflow changes, due to climate variation and human influence, in the important Karkheh River Basin in western Iran. To validate the results, hydrological modeling (HBV model) and Landsat 5 Thematic Mapper (TM) images were used for the study period between 1980 and 2012. The recently developed DBEST (Detecting Breakpoints and Estimating Segments in Trend) method identified an abrupt negative change in the streamflow trend in 1994–5. The results show that the observed streamflow decrease in the Karkheh River is associated with both climate variation and human influence. The combination of increased irrigated area (from 9 to 19% of the total basin area), reduction of forests (from 11 to 3%), and decreasing annual precipitation has significantly reduced streamflow in the basin. Moreover, the results show that the streamflow reduction in the Karkheh Basin is more sensitive to the change in precipitation than temperature.

## 1 Introduction

The Karkheh River Basin (KRB), called "the food basket of Iran", is one of the most important agricultural basins in Iran. Irrigated farmland in the basin produces wheat for the entire country, while non-irrigated areas yield grain and livestock products (Ahmad & Giordano, 2010). The KRB is equally essential for hydropower purposes. Nonetheless, due to frequent droughts, massive agricultural activities, and dam construction programs, the KRB has experienced significant streamflow reduction in recent decades (Ahmad & Giordano, 2010). If the streamflow continues to decrease in the near future, the sustainability of food production may be jeopardized. Therefore, it is of vital importance to investigate the reasons for the streamflow reduction and develop a management plan (Masih et al., 2011).

Climate variation affects the streamflow directly. For example, changing precipitation patterns and intensity, together with changing temperatures, will greatly modify the streamflow (Jones et al., 2009; Schaake, 1990; Teng et al., 2012). On the other hand, human activities such as land use change, water withdrawal, and hydraulic structures have substantial impacts on streamflow (Guyennon et al., 2017; Haddeland et al., 2014). A basin-wide investigation to quantify the contribution of climate





variation and human activities on streamflow change means a better understanding of spatial effects on the flow regime. This

is useful for water professionals to help them implement proper plans for water management at the basin-scale (Liu et al., 2017; Teng et al., 2012).

Hydrological modelling and the Budyko method are widely used to quantify the contribution of human influence and climate variation to streamflow change (e.g., Chiew et al., 2009; Fu et al., 2007; Hu et al., 2012; Sankarasubramanian et al., 2001; Teng et al., 2012; Hashemi et al., 2015). Wang et al. (2013) used a simple water balance hydrological model and the Budyko

method to analyze streamflow reduction in the Haihe River in China during the 1957–2000 period. As a first step, they used the Mann-Kendall test to investigate the temporal trend of streamflow and detected a breakpoint in the annual streamflow trend that divided the study period into a pre- and post-change period. They were able to quantify the contribution of climate variation and human activities on the Haihe streamflow reduction, assuming that only climate variation and human activities cause streamflow changes and that these are independent parameters. Their results indicated that human activities, such as the

expansion of agricultural areas, dam construction, and industrial activities, were responsible for more than 50% of the streamflow reduction in the basin (Wang et al., 2013). Geris et al. (2015) employed the HBV model to investigate the flow regime variation (i.e., magnitude, frequency, and duration) in a highly regulated catchment in the River Lyon, Scotland. The aim was to develop a model applicable to highly regulated areas and data-sparse regions. The results indicated that the flow regime had reduced in magnitude, frequency, and duration, at both inter- and intra-annual scales, due to regulation activities

in the catchment (Geris et al., 2015). In a study conducted by Birhanu et al. (2019), the HBV model was applied to the Gumara catchment's water system in Ethiopia to assess the impact of land use and land cover change on the water cycle and, more specifically, streamflow variation. Three years of Landsat images from 1986, 2001, and 2015 were used to analyze the impact of human activities, such as deforestation and cultivation. However, the results showed very little change in catchment runoff despite a massive change in land use. The uncertainty in modelling, parameterization, and assessing land use/land cover

representation, might have affected the analysis and modelling outcome (Birhanu et al., 2019).

In another study, the Budyko method was used to investigate the cause of streamflow change in the South Atlantic basins (Patterson et al., 2013). Their findings showed that climate change tended to increase the streamflow, while human activities, such as agricultural land expansion, masked the climate impact and reduced the streamflow. Since human activities can exacerbate or reduce the impacts of climate change, their study highlights the importance of assessing climate variation impacts

as well as studying the influence of local drivers (Patterson et al., 2013). Liu et al. (2017) studied streamflow variation in ten river basins at different locations in China using the Budyko method. One of the main goals was to investigate streamflow sensitivity due to human influences such as population increase, economic development, and water reservoirs in the basins. The results suggest that in comparison to the south, basins in northern China are more sensitive to climate variation, which makes water resource management in the northern basins a more challenging task. They found that, in recent years, human

activities have become a major driver affecting streamflow, which adds uncertainty to management plans at both spatial and temporal scales (Liu et al., 2017).





To assess the hydrological response of a basin at daily or monthly time-scale, hydrological modelling can be a useful method. The Budyko method, on the other hand, can be used to analyze the responses of watersheds to climatic variation in a more straightforward and systematic approach (Wu et al., 2017). This method has proven to be a useful technique for extended time

periods and, unlike hydrological models, does not require detailed parameterization or complex data sets (e.g., Dooge, 1992; Hu et al., 2012; Huang et al., 2016; Li et al., 2012; Liu et al., 2017; Teng et al., 2012; Xu et al., 2014a; Zeng et al., 2015; Zheng et al., 2009). However, when the period of study is sufficiently long, the Budyko method assumes the water cycle to be static and water storage change is believed to be negligible (Liu et al., 2017).

In view of the above, the current paper investigates the performance of a hydrological model (the HBV model) and the Budyko

method in analyzing the mean annual streamflow variation of the KRB in western Iran during the last three decades (from 1980 to 2012). The results from the Budyko method were compared with the HBV model. To the authors' knowledge, this is the first attempt to use the Budyko method to analyze streamflow variation in a major river basin in Iran. For further validation, the remote sensing technique was employed to provide multiple land use maps of the basin area for the study period. This type of validation is addressed less in the literature for major basins like the KRB, where data scarcity is a general problem for

longer time periods. Trend analysis was carried out with the newly developed detecting breakpoints and estimating segments in trend (DBEST) method (Jamali et al., 2015). One of the advantages of this method is that it detects different types of patterns and trends (and their significance) in streamflow variations. Consequently, the main objectives of this study were i) to quantify the impact of human activities and climate variability on streamflow reduction in the KRB, ii) to compare and analyze the results from the Budyko method with the HBV model, and iii) to develop land use maps employing remote sensing techniques

for the validation of the outcomes of the applied methods.

## 2 Materials and Methods

### 2.1 Study area

The KRB, with an area of 43,000 km$^2$ stretching over seven provinces and 32 districts, is located in the western part of Iran between 30° and 35° N latitude and 46° and 49° E longitude (Fig. 1). The basin is one of the primary sources of wheat

production in Iran and encompasses 9% of the total irrigated area of the country (Muthuwatta et al., 2010). Five major rivers flow through the KRB, and the basin is divided into five main sub-basins named Gamasiab, Qarasou, Kashkan, Seimareh, and Upper Karkheh (Ahmad & Giordano, 2010). The basin is relatively dense, accommodating 5% of Iran's population, which makes it the third most populated basin in the country. The KRB has 3.5 million residents, from which 40% live in urban areas. The population is mostly concentrated in Qarasou sub-basin, which occupies 17% of the KRB area and houses 30% of the

population. The southern regions of the basin, with almost 11% of the area, are home to 5% of the population (JAMAB consulting, 1999; JAMAB consulting, 2006; Marjanizadeh et al., 2010).

Figure 1 illustrates the location of the KRB in Iran and its sub-basins. The Upper Karkheh sub-basin is located above the Karkheh dam. The flow that reaches the Upper Karkheh outlet is drained from the entire study area.



**Figure 1: Location of the KRB in Iran. ET stands for evapotranspiration.**

The altitude of the basin varies from less than 10 m in the south to more than 3,500 m above mean sea level in the north. Annual precipitation ranges from 150 mm in the southern parts to 750 mm in the northern regions (see Table 1). The maximum summer temperature varies between 35 and 45° C across the basin. Rangeland, rainfed agriculture, forest, and irrigated
agriculture are the dominant land uses (Muthuwatta et al., 2010).

**Table 1: Karkheh sub-basins' characteristics.**

Springwater is traditionally used for irrigation, however, due to frequent droughts and subsequent surface water scarcity, the
pumping of groundwater and river water diversions have become widespread in recent decades. Competition between irrigated agriculture and the need for wetland ecosystems has led to low water productivity, increasing salinity, and reduced surface water availability in the lower parts of the basin (Muthuwatta et al., 2010).

During the twentieth century, the KRB remained mainly unregulated. The first dam constructed in the area was the Karkheh dam, which was completed in 2001 and was the first large multipurpose dam in Iran, with a total storage of 5,600 MCM (Table
2). Its reservoir is designed to irrigate 320,000 ha of agricultural land in the Upper Karkheh basin. The Seimareh dam was built in 2013 and there are a few smaller reservoirs under operation with several other dams and irrigation schemes either under construction or under planning, all of which could put additional strain on the streamflow of the study area (Muthuwatta et al., 2010).

**Table 2: KRB's major dams.**

**2.2 Data**

The study period (1980–2012) was selected considering the availability and quality of data. Daily precipitation data for the study period were acquired from a well-distributed gauge network across the study area. The Thiessen method was used to determine the weight of each station for the total precipitation of the sub-basins.
The potential evapotranspiration ($E_0$) stations are not spatially representative of the basin, though these stations (hereafter called reference stations) have a sufficiently long period of recorded temperature and $E_0$. The temperature stations, however, are relatively well-distributed in the basin and provided a long period of recorded temperature, but no $E_0$ information. Therefore, temperature data were used to estimate spatially distributed $E_0$. The temperature stations were classified based on their altitude and their distance from the reference stations (Lambert and Chitrakar, 1989). Accordingly, for each year, a
monthly relationship between observed $E_0$ and temperature in the reference stations was developed. Next, temperature data





were used to calculate $E_0$. For instance, in 1995, for the Polchehr station, which is one of the reference stations (Fig. 1), there is a good correlation between monthly $E_0$ and monthly mean temperature ($R^2 = 0.95$) (see Fig. 2):

$$E_{0(P)} = 15.103 \cdot T_{(P)} - 20.929 \qquad\qquad (1)$$


where T is temperature. Given that the Kermanshah temperature station is located at the same altitude and has a significant correlation with Polchehr station (i.e. 0.94), $T_{Kermanshah}$ was substituted in Eqn. (1) to estimate $E_0$ for the Kermanshah station.

**Figure 2: $R^2$ between monthly evaporation ($E_0$) and temperature (T) for the Polchehr reference station in 1995.**


Similarly, for each year of the study period and each temperature station, a relationship was derived to provide spatially distributed $E_0$ data. Table 3 shows the temperature stations that were correlated with the $E_0$ reference stations.

**Table 3: $E_0$ reference and related temperature stations.**


Regarding streamflow data, five discharge stations located at the outlet of the sub-basins, namely Polchehr at Gamasiab River, Ghoorbaghestan at Qarasou River, Poldokhtar at Kashkan River, Nazarabad at Seimareh River, and Payepol at Upper Karkheh River, were selected considering their locations, duration of records, and quality of data (Fig. 1 and Table 4). The Payepol station is located downstream of the Karkheh dam and receives a cumulative discharge from the upstream sub-basins. Thus, it

can provide useful information about the impact of the reservoir on the main river flow.

**Table 4: Streamflow stations and characteristics.**

To calculate the equivalent depth of streamflow of each sub-basin in millimetres (as presented in Table 4), the mean annual

flow at the outlet of the discharge station was divided by the area of the sub-basin. However, as the Seimareh sub-basin receives a part of its streamflow from the two upper sub-basins (i.e., Qarasou and Gamasiab), the mean annual flow was divided by the summation of the three sub-basins. Likewise, for the Upper Karkheh sub-basin, which receives water from the upper sub-basins (Masih et al., 2011), the entire drainage area was considered while calculating the mean annual flow.

### 2.3 Methodology

Two approaches, HBV modelling and the Budyko method, were employed to study the impacts of climate variation and human activities on mean annual streamflow change in the KRB. The first step before applying the methods was to analyze the streamflow trend in order to pinpoint any changes or breakpoints in the mean annual streamflow during the study period. Major land use change in a basin can result in a gradual or abrupt streamflow change (Li et al. 2007). The DBEST method was applied



to detect a change in streamflow trends. DBEST is a user-friendly program for analyzing time-series data with two main
applications of generalizing trends in main features and detecting and characterizing trend changes. It uses a novel
segmentation algorithm that simplifies the trend into linear segments, using the number of changes or a threshold for the
magnitude of changes of interest for detection. In addition to detecting trend changes and calculating the statistical significance
of the trend, DBEST determines the timing, magnitude, number, direction, and type (abrupt or gradual) of the detected changes
(Dey and Mishra, 2017; Jamali et al., 2015).

The detected breakpoint divides the study period into a pre- and post-change period. It is assumed that the streamflow was not
influenced by human activities before the breakpoint (the pre-change period), and any flow changes were due to climate
variation. For the period after the breakpoint (the post-change period), both climate variation and human activities affect the
streamflow (Dey and Mishra, 2017).

**2.4 Assessment of streamflow changes using the HBV hydrological model**

HBV is a conceptual semi-distributed rainfall runoff model, developed by the Swedish Meteorological and Hydrological
Institute, that uses observed daily precipitation, air temperature, vapor pressure, and wind speed as inputs to simulate daily
discharge. The HBV model was selected due to its simple yet flexible structure. This is an important feature for a model to
simulate a basin like Karkheh, which covers a large space from high mountainous terrain to low land areas at sea level. In the
model, the basin area is subdivided into different elevations and vegetation zones. The HBV model has been successfully
applied in many areas, including snow-influence areas as well as semiarid climates in both local and regional studies (e.g. Al-
Safi et al., 2019; Al-Safi & Sarukkalige, 2017a; Al-Safi & Sarukkalige, 2017b; Al-Safi & Sarukkalige, 2018; Götzinger &
Bárdossy, 2007; Li & Zhou, 2016; Li et al., 2016; Lidén & Harlin, 2000; Lindström et al., 1997; Love et al., 2010; Merz &
Blöschl, 2004).

The HBV model uses three routines for simulation, a precipitation routine, a soil moisture routine, and a response routine. The
calibration parameters for the model are beta, fc, lp, athorn, cflux, perc, khq, k4, maxbaz, cfmax, dttm, pcorr, rfcf, sfcf, and tt.
The input data for the soil moisture routine are fc, β, and lp. Parameter fc is the maximum capacity of soil moisture storage.
Parameter β regulates any increase in soil moisture storage following a millimetre of precipitation. The parameter lp refers to
the value of soil moisture above which actual evapotranspiration is equal to potential evapotranspiration. athorn refers to the
calculation of evapotranspiration. Capillary flow is controlled by cflux. The parameters perc, khq, and k4 control the response
routine, which both manages water extracted from soil moisture to runoff and controls the precipitation and evapotranspiration
that affect wet areas such as lakes and rivers. The maxbaz parameter refers to the transformation function, which calculates
outflow from the basin. The cfmax, dttm, and tt parameters are responsible for snowmelt and are used to calculate the melting
factor. pcorr, rfcf, and sfcf are correction factors and control the total volume (Al-Safi et al., 2019; Al-Safi & Sarukkalige,
2018; Kazemi et al., 2019; Lindström et al., 1997).

In order to employ the HBV model for the quantification of the contribution of climate variation and human activities on
streamflow change, the first step is to calibrate and validate the model for the pre-change period. For the next step, adopting



the same parameters, HBV is used to simulate the flow for the post-change period (Chang et al., 2016). Accordingly, $\Delta Q_c$ (i.e., streamflow change due to climate variation) is calculated by deducting the mean annual simulated streamflow from the post-change period and that of the pre-change period. $\Delta Q_H$ (i.e., streamflow variation due to human activities) is calculated as the

difference between the mean annual simulated and observed streamflow, both from the post-change period (Chang et al., 2016; Hu et al., 2012; Sun et al., 2014).

Two evaluation indices, $R^2$ and accumulated difference ($\delta$) (Eqn. [2] and [3]), were employed to evaluate the calibration procedure (Al-Safi et al., 2019):


$$R^2 = \frac{\Sigma(Q_o - \overline{Q_o})^2 - \Sigma(Q_s - Q_o)^2}{\Sigma(Q_o - \overline{Q_o})^2} \tag{2}$$

$$\delta = \Sigma(Q_s - Q_o) \cdot C_t \tag{3}$$

where $Q_s$ is HBV simulated streamflow, $Q_o$ is observed streamflow, $\overline{Q_o}$ is the mean annual streamflow, $C$ is a coefficient

transforming units to mm over the basin, and t is time.

## 2.5 Assessment of streamflow changes using the Budyko method

The Budyko method defines a physically understandable link between annual evapotranspiration and average water and energy balance at the basin level. Equation (4) was proposed by Budyko (1974) to display the relationship between actual and potential

evapotranspiration (McMahon et al., 2013; Wang et al., 2016; Sposito, 2017):

$$\frac{E}{P} = \left[\frac{E_0}{P}\tanh\left(\frac{E_0}{P}\right)^{-1}\left(1 - \exp\left(\frac{E_0}{P}\right)\right)\right]^{0.5} \tag{4}$$

where $E_0$ is the potential evapotranspiration [mm day$^{-1}$], $E$ is actual evapotranspiration [mm day$^{-1}$], and $P$ is precipitation [mm day$^{-1}$]. The method was further extended by Fu (Eqn. [5]) and Choudhury (Eqn. [6]) to calculate the effects of climate variation

and human impact on streamflow at the basin level (Wang et al., 2016). In these two equations, a new element is introduced called catchment characteristic parameter (ω in the Fu equation and n in the Choudhury equation) (Wang et al., 2016). The catchment characteristics are introduced as empirical parameters, which represent soil properties, slope, land use, and climate seasonality (Al-Safi et al., 2019; Li et al., 2013; Liang et al., 2015).





$\quad \frac{E}{P} = 1 + \frac{E_0}{P} - \left[ 1 + \left( \frac{E_0}{P} \right)^\omega \right]^{\frac{1}{\omega}}$ (5)

$\frac{E}{P} = \frac{1}{\left( 1 + \left( \frac{P}{E_0} \right)^n \right)^{1/n}}$ (6)

The catchment characteristic parameter is derived by minimizing the objective function (obj) stated in Eqn. (7) (Li et al., 2013; Al-Safi et al., 2019; Kazemi et al., 2019). A higher n or ω means a higher ET for a given P and $E_0$, and therefore lower

streamflow (Q) (Xu et al., 2014b):

$obj = min \sum_i \left\{ \frac{Ei}{Pi} - \left\{ \frac{1}{\left( 1 + \left( \frac{Pi}{E_0 i} \right)^n \right)^{1/n}} \right\} \right\}^2$ (7)

The Budyko method assumes that water storage change (ΔS) is insignificant over a sufficiently long period. Therefore, Eqn. (8), can be rewritten in the form of Eqn. (9) (Xu et al. 2013). Equation (10) shows the total variation of streamflow (ΔQ):


$\bar{P} = \bar{E} + \bar{Q} + \Delta S$ (8)

$E = P - Q$ (9)

$\Delta Q = \overline{Q_{o2}} - \overline{Q_{o1}}$ (10)

where numbers 1 and 2 denote the periods of study.

The change in streamflow (ΔQ) is believed to be a combination of changes due to climate variation ($\Delta Q_c$) and human activity ($\Delta Q_{HA}$) (Al-Safi et al., 2019; Kazemi et al., 2019, Li et al., 2012; Liang et al., 2015).

$\Delta Q = \Delta Qc + \Delta Q_H$ (11)


The analytical elasticity method was employed to define the contribution of each of the two variables influencing the streamflow. Equation (12) was developed based on the Budyko equation (Schaake, 1990; Yang & Yang, 2011):

$\varepsilon_P(P, Q) = \frac{dQ}{dP} \cdot \frac{P}{Q}$ (12)

The equation further was improved by Yang and Yang (2011) to the present form (Eqn. 14 and 15). In this method, $\varepsilon_P$ and

$\varepsilon_{E_0}$ are assumed to be independent variables. Equation (13) defines the streamflow change influenced by climate variation. $\varepsilon_P$ and $\varepsilon_{E_0}$ in Eqn. (14) and (15) are precipitation elasticity and potential evapotranspiration elasticity, respectively. Finally, Eqn. (16) suggests the impact of human activity on streamflow variation (Liang et al., 2015; Yang & Yang, 2011):



$$\Delta Q_C = \varepsilon_P \frac{\Delta P}{P} \bar{Q} + \varepsilon_{ET_0} \frac{\Delta E_0}{E_0} \bar{Q} \tag{13}$$


$$\varepsilon_P = \left\{ 1 - 1 / \left[ 1 + (\tfrac{P}{E_0})^n \right]^{1+1/n} \right\} / \left\{ 1 - 1 / \left[ 1 + (\tfrac{P}{E_0})^n \right]^{1/n} \right\} \tag{14}$$

$$\varepsilon_{E_0} = - \frac{1}{\left[ 1 + \left( \frac{E_0}{P} \right)^n \right]^{1+\frac{1}{n}}} \cdot \frac{1}{\frac{1}{E_0/P} \frac{1}{\left[ 1 + \left( \frac{E_0}{P} \right)^n \right]^{\frac{1}{n}}}} \tag{15}$$

$$\Delta Q_H = \Delta Q - \Delta Q_C \tag{16}$$

## 2.6 Analyzing Land Use–Land Cover (LULC) change during the study period

Developing water distribution systems and the construction of reservoirs in the KRB have improved food production and provided easier lives for the residents of the area. However, it has disturbed the water cycle in the basin. The KRB is considered

a semiarid to arid region, which is vulnerable to water scarcity. Increasing human population, agricultural and industrial activities, and urbanization demands, combined with the governmental policy of being self-sufficient in agriculture, have put additional pressure on water resources in the area (Ahmad et al., 2009; Masih, 2011). Any change in LULC (land use–land cover) can change the water balances by modifying, for example, groundwater storage, soil infiltration, and actual evapotranspiration (Mekonnen et al., 2018). Therefore, given the scarcity of land cover information in the KRB, Multispectral

Landsat satellite imagery was used to investigate the probable relationships between the land cover change and streamflow variation in the basin. Multispectral Landsat satellite is a remote sensing approach that provides spatio-temporal information of the study area over time (Ghobadi et al., 2012; Jafari & Hasheminasab, 2017; Muttitanon & Tripathi, 2005). Landsat 5 Thematic Mapper (TM) was selected for this study because it offers high-resolution images (120 m) and complete spatial coverage of the study basin from 1980 to 2012.

Cloud-free Landsat images were obtained from Landsat 4-5 TM C1 Level-1 for the three years, 1987, 1995, and 2012. These images were downloaded from the United States Geological Survey official website (http://earthexplorer.usgs.gov/) and were projected to the UTM (zone 38) and WGS 84 datum reference system. Also, the ground truth data were collected using the Global Positioning System (GPS) and ground control points from the Google Earth application to provide a signature for each land use type. These data were applied for classification and overall accuracy assessment of the classified images. The image

classification processing was performed in ENVI 4.8 Environment, employing a supervised classification technique with the maximum likelihood algorithm for generating the land use map. The maximum likelihood classifier (MLC) is the most common statistical technique for image classification and for evaluating the standard LULC (Zaidi et al., 2017). Due to the complexity of the land use types in the basin, as well as overlaps among different land use types, an acceptable threshold was



determined to simplify the classification of the study area (Figure 9). Eventually, five types of land use were detected in the
study area: irrigated (merged with about 5–10% pasture area), rainfed (merged with about 10–20% irrigated and pasture area),
range and pasture, forest (merged with about 10–20% pasture area), and urban (includes buildings and orchards).

**2.7 Uncertainty analysis**

In order to adequately simulate a hydrological response at the basin level, accurate data such as climate variables (precipitation,
ET, etc.) and catchment physical characteristics (topography, land coverage, vegetation, etc.) are vital. In climate variation
related studies, in which the study period is on the scale of decades, it is difficult to gather uniformly distributed and accurate
data sets. Therefore, understanding and quantifying the main sources of uncertainty help to give a better understanding of the
results and improve the related management decisions (Kapangaziwiri et al., 2009).

To address this issue, climate elasticity of streamflow to precipitation and evapotranspiration is suggested (Eqn. 12) (Schaake,
1990; Yang & Yang, 2011, Yang et al., 2014). The equation can be rewritten to the form of Eqn. (17) (Yang et al., 2014) to
derive the possible error of estimating streamflow due to climate variation.

$$dQ = \varepsilon_a \frac{da}{a} \cdot Q \tag{17}$$

where $Q$ is flowrate (mm/year), $\varepsilon$ is streamflow elasticity, and $a$ is a climate parameter.
In the present study, after defining the main possible sources of error, an uncertainty analysis was carried out to find the
sensitivity of streamflow to each of the parameters.

**3 Results**

**3.1 Streamflow analysis**

The DBEST method was applied to analyze streamflow trends in the study area. For each sub-basin, DBEST detected a
breakpoint in the streamflow time series with a statistical significance of 95%. Abrupt changes were detected for all the sub-
basins, which mostly occurred around 1994–95. To prevent overloading the paper with figures and diagrams, the Qarasou
streamflow is shown as an example, in Figure 3.

**Figure 3: Streamflow trend in the Qarasou sub-basin and the breakpoint detected by DBEST.**

In this sub-basin, the annual streamflow experienced a dramatic decrease from the pre-change period (1980-1994) to the post-
change period (1995–2012). Table 5 illustrates the breakpoints for all sub-basins, as well as the quantified change in
streamflow.





**Table 5: Average discharge variations from the pre-change period ($Q_1$) to the post-change period ($Q_2$) and the magnitude of the change.**

Further investigation of streamflow showed a substantial difference between the flow duration curves (FDCs) for the pre- and post-change periods (Fig. 4). For instance, Figure 4 shows for the Payepol station that the mean daily streamflow decreased

by a third, during 50% of the time. The mean monthly flows in the sub-basins also experienced major reductions for almost every month (Fig. 5), with the exception of the Payepol station in which the streamflow showed an increase in the summer months (June, July, and August) in the post-change period compared to the pre-change period. This might be the result of river flow regulation due to the operation of the Karkheh Dam since 2001. The reservoir water, which is stored during abundance, is released in the summer season. Therefore, an increase in summer flow does not necessarily mean an increase in natural

streamflow from the basin. Moreover, Figure 5 shows that for the Kashkan sub-basin, the difference between stream flows of the two periods in late autumn and winter (i.e., October, November, December, January, and February) is less than other sub-basins, while it shows largest differences for the summer months (i.e., June, July, and August). It can be postulated that intensive agricultural activities in the area have led to higher actual evapotranspiration and, therefore, the larger difference between stream flows of the two periods.


**Figure 4: FDCs of Karkheh sub-basins for the pre-change (1) and post-change (2) periods.**

**Figure 5: Mean monthly streamflow of the Karkheh sub-basins during the pre-change (1) and post-change periods (2).**

As presented in Table 6, almost all of the sub-basins experienced a reduction in precipitation and an increase in evapotranspiration from the pre-change to the post-change period. The Qarasou sub-basin showed the most severe decline in average annual precipitation (-19%) but the lowest change in average potential evapotranspiration rate (+5%). On the other hand, the Kashkan sub-basin experienced almost no change in precipitation (+1%) but the largest increase in potential evapotranspiration (+18%).


**Table 6: Climate variation during pre- and post-change periods. Numbers 1 and 2 denote the pre- and post-change period, respectively.**

### 3.2 Hydrological modelling

After establishing the breakpoints in the streamflow time series using the DBEST method, the HBV model was calibrated for

the pre-change period and all sub-basins. The suggested and calibrated ranges of parameters for the sub-basins are presented in Table 7. Later, the post-change period was simulated using the parameters calibrated for the pre-change period (Fig. 6).



**Table 7: Suggested range for the calibration parameters of HBV.**

The annual streamflow depicted in Figure 6 and the evaluation indices presented in Table 8 imply that the HBV model was well calibrated for the pre-change period with the weakest performance observed for the Kashkan sub-basin ($R^2 = 0.66$). All evaluation indices in Table 8 are based on a daily time scale, but Figure 6 shows the annual scale for the sake of presentation.

**Figure 6: Simulated vs. observed annual streamflow for the KRB.**


The HBV model overestimated the streamflow for the post-change period, which suggests that there existed factors other than climate variations affecting the streamflow of the study basins. These factors are believed to be related to human activities (Hu et al., 2012).

**Table 8: Evaluation index ($R^2$) for the sub-basins.**

**3.3 Budyko method**

The parameters of precipitation ($\varepsilon_P$) and evapotranspiration elasticity ($\varepsilon_{E0}$) were calculated using Eqn. (14) and (15) (Table 9). The higher values of $\varepsilon_P$ in comparison to $\varepsilon_{E0}$ for all sub-basins suggest that the hydrological behaviors of the sub-basins are more sensitive to variation in precipitation than evapotranspiration. The negative $\varepsilon_{E0}$ indicates that evapotranspiration and

streamflow are inversely related. As can be seen in Table 9, the $\Delta Q_c$ and $\Delta Q_H$ from the HBV model and the Budyko method vary between the sub-basins, but for any given sub-basin the results are compatible.

**Table 9: Comparison between HBV model and Budyko method for estimation of streamflow changes in the Karkheh sub-basins.**

Figures 7 and 8 provide a visual comparison between the used methods. As mentioned earlier, Kashkan yielded the weakest performance among the sub-basins regarding the HBV modeling, which can be a result of data quality.

**Figure 7: Streamflow changes for all sub-basins due to climate variability and human activities. CV and HA denote climate change and human activities, respectively.**


Results for the Seimareh sub-basin are the cumulative response of the three sub-basins Qarasou, Gamasiab, and Seimareh (Fig. 7). These were affected mostly by climate rather than human activities. In the Upper Karkheh, which receives the cumulative responses of all sub-basins, both drivers (CV and HA) were almost equally responsible for streamflow decline (Fig. 8).




## 3.4 Land use change

Using the information from remote sensing, a better sense of the human influence leading to land use change can be provided. The land cover change during the study period was investigated using Landsat 5 TM as it is the only satellite mission that provides images dating back to the 1980s. Three years 1987, 1995, and 2012 were selected to present the land use condition of KRB. These three years represent the KRB land use condition for three phases of before breakpoint, breakpoint, and after breakpoint, respectively.

Figure 9 shows a noticeable expansion of irrigated farmlands in the basin during the three investigated years. Before the breakpoint, the majority of rainfed and rangelands were located in the mountainous region of the basin. Forests mainly covered the middle and south eastern parts of the basin while irrigated areas were scattered in the northern parts (Fig. 9 [a]). However, by 2012, as presented in Figure 9 (c), irrigated farms spread throughout the basin.

**Figure 9: Land use maps for the KRB, (a) before breakpoint (1987), (b) breakpoint (1995), and (c) after breakpoint (2012).**

Five major LULC classes were identified in the study basin, namely irrigated, rainfed, range and pasture, forest, and urban areas. The LULC maps show that the KRB is covered predominantly by rangeland and rainfed farms. Although the basin is still covered mostly by these land use types, the area of the classes has changed since the 1980s (Table 10).

Tables 6 and 10 show that the land use change (~70% reduction in dense forest area and 100% increase in irrigated farms), combined with a 13% decline in rainfall, led to more than 40% streamflow reduction for the entire basin. Thus, streamflow reduction is due to both climate variation and human activities in the KRB. The present study suggests that all sub-basins in the KRB experienced a major abrupt change of streamflow during the 1994–1995 period, which coincides with the period of dam construction in the basin.

**Table 10: Percentage of land use class change in five different sub-basins.**

**Figure 10: From left to right, KRB land use classification before breakpoint (1987), during the change (1995), and after breakpoint (2012).**

**Table 11: Land use classes (total) percentage in different years.**

Figure 10 and Table 11 show that the forest area significantly decreased from 1987 to 2012, while the size of both irrigated areas and urban areas increased noticeably during the study period. Deforestation occurred in the southeastern part of the basin (i.e., in the Kashkan sub-basin). The calculated land use map, together with the results from the Budyko and HBV methods, suggests that the streamflow decrease in Kashkan sub-basin is mainly related to human activities rather than climate variation.



In the case of Upper Karkheh sub-basin, in addition to the cumulative response of the upper sub-basins, the Karkheh dam plays

a major role in streamflow reduction. Based on a study conducted by Ahmad et al. (2010), actual annual evapotranspiration varies from 41 to 1,681 mm/year throughout the basin, with the highest rate for the Karkheh dam (Ahmad & Giordano, 2010), which indicates the substantial impact of the Karkheh dam on water balance.

### 3.5 Uncertainty analysis

The input data and the model structure are considered the main sources of uncertainty in hydrological studies (Kapangaziwiri

et al., 2009). Hence, in the applied Budyko method, potential evapotranspiration and precipitation as the main input data can be recognized as the possible sources of uncertainty. Precipitation was calculated by taking the average precipitation of all the stations in the sub-basins, which is, basically, a simplification of the physical characteristics of the input water to the system. Due to the lack of uniformly distributed stations, a regression method was used for the case of evapotranspiration data, which again can introduce error to the modelling result. For the case of the Budyko method, the catchment characteristic parameter

($n$) was assumed to be constant during the study period. However, the value of $n$ is dependent on other climate parameters, such as climate seasonality, mean storm depth (Yang et al., 2014), vegetation coverage (Li et al., 2013; Yang et al., 2009), and/or effective rooting depth and plant root characteristics (Cong et al., 2015; Donohue et al., 2012).

Table 12 presents the possible sources of error in streamflow estimation, as a result of a 10% error in Budyko parameters (i.e., P, $E_0$, n), for each sub-basin. For instance, the results suggest that a 10% error in precipitation leads to a 3–5% error in

streamflow estimation. On the other hand, the results indicate that the streamflow is not as sensitive to the possible error in evapotranspiration data.

**Table 12: Streamflow response to a 10% error in the Budyko input parameters.**

Figure 12 suggests a relationship between $\varepsilon_p$, $\varepsilon_{E_0}$, and $n$ and the aridity index ($E_0$/P). It also explains the response of Budyko equation curves with regards to the variation of $n$. Using Figure 12 and equation 14, Table 12 displays the change in streamflow, in the case of a 10% error in the value of $n$.

**Figure 12: Budyko equation curve responding to different values of $n$.**


### 4 Discussion

The KRB plays a key role in food production in Iran. Thus, any hydrological change in the basin directly affects the livelihoods of farmers as well as urban consumers at both the basin and country levels. Figure 11 shows the rural and urban populations of each sub-basin.






**Figure 11: Population distribution in the KRB according to 2015 national records.**

Previous research for the basin suggests that, because of agricultural demand and recurrent droughts, the basin has experienced the over-exploitation of groundwater and increasing pressure on surface water (Ahmad & Giordano, 2010). Based on a

conversation with the former head of the agricultural organization of Lorestan province, in which the Kashkan sub-basin is located, there was a substantial investment in expanding agricultural lands in this catchment during the 1990s. The primary purpose of this investment was to create jobs for locals. Kashkan, with a rural population of almost 300,000, is the most densely populated sub-basin in the Karkheh catchment. The population density and development of agricultural lands are in line with the result of this paper, suggesting that human influence is a dominant factor in streamflow reduction in this specific catchment.

The Seimareh sub-basin also experienced extensive agricultural development during the study period. Therefore, as presented in Table 10, the agricultural land cover change in Seimareh and Kashkan sub-basins is higher when compared to other sub-basins.

The population is predicted to increase to 4.8 million by the year 2025, out of which 75% will live in urban areas. Urbanization in the northern sub-basins will increase the strain on water resources in the basin, while in the southern part of the basin, dam

constructions and 300,000 ha planed irrigation programs would be overwhelming for the groundwater and surface water (Marjanizadeh et al., 2010). Accordingly, policymakers should carefully follow any change in the quality and quantity of the available freshwater and changes in the hydrological behavior of the streamflow.

**5 Conclusion**

The Budyko method and hydrological modelling were employed to analyze the impacts of climate variation and human

activities on streamflow decrease in the important Karkheh River Basin in Iran. Streamflow analysis suggested a notable reduction after the 1994–1995 period. Our results show that for most of the sub-basins, climate variation and human activities (i.e., agriculture, deforestation, and water diversion) are more or less equally responsible for the streamflow reduction. Land use maps were provided using Landsat 5 TM images to verify the hydrological assessment, which suggested significant changes in land cover throughout the basin for the study period between 1980 and 2012.  The Budyko method showed to be a

reliable, easy to implement, user-friendly method to analyze streamflow changes during the study period.

The outcome of this study can be used to assist policymakers and water professionals in proposing a proper water management plan to prevent the further reduction of streamflow and groundwater storage. It can also help to offer appropriate urban and rural development plans, ecological restoration, conservation projects, regulation of water for irrigation, and sustainable management of the KRB in the future. In the case that the current water management and ecosystem planning remains

unchanged, the impacts of climate variation and human influence may severely damage the stability of the ecosystem of the entire basin.





**Data availability**

The hydrological data were acquired from the regional water companies of the corresponding provinces, as well as the Iran Water Resources Engineering Company. The meteorological data were provided by the Iran Meteorological Organization.

**Author contribution**

Hamideh Kazemi: Conceptualization, Data curation, Formal analysis, Investigation, Methodology, Software, Validation, Visualization and Writing (original draft), Writing (reviewing and editing)

Hossein Hashemi: Funding acquisition, Methodology, Project administration, Resources, Supervision, Validation and Writing (reviewing and editing)

Fatemeh Fadia Maghsood: Software, Writing (reviewing and editing)

Seyyed Hasan Hosseini: Data curation, Methodology, Writing (reviewing and editing)

Ranjan Sarukkalige: Supervision and Writing (reviewing and editing)

Sadegh Jamali: Software, Writing (reviewing and editing)

Ronny Berndtsson: Funding acquisition, Supervision, Validation and Writing (reviewing and editing)

**Acknowledgment**

This project was funded in part by the MECW project at the Centre for Middle Eastern Studies, Lund University, Sweden.

**Conflict of interest statement**

The authors declare no conflicts of interest.





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



## Tables

**Table 1: Karkheh sub-basins' characteristics**

| Sub-basin | Area (km$^2$) | Mean altitude (m amsl) | Mean precipitation (mm/year) | Mean streamflow (mm/year) |
|---|---|---|---|---|
| Qarasou | 5,508 | 1,559 | 424 | 111 |
| Gamasiab | 11,512 | 1,856 | 461 | 81 |
| Kashkan | 9,524 | 1,611 | 477 | 158 |
| Seimareh | 12,350 | 1,179 | 412 | 97 |
| Upper Karkheh | 3,995 | 795 | 422 | 114 |

**Table 2: The KRB's major dams**

| Name | Longitude ºE | Latitude ºN | Storage-normal level (BCM) | Purpose | Operation date |
|---|---|---|---|---|---|
| Karkheh | 48.1506 | 32.4208 | 5.6 | Irrigation, hydropower generation, flood control | 2001 |
| Seimareh | 47.1908 | 33.3183 | 2.8 | hydropower generation | 2013 |

**Table 3: E0 reference stations and related temperature stations**

| Reference E0 station | Altit. (m) | Temp. station | Altitude | Altit. (m) |
|---|---|---|---|---|
| Chamanjir | 1140 | Khorramabad | | 1147 |
| | | Koohdasht | | 1190 |
| Dasht Abbas | 161 | Dehloran | | 232 |
| Abdolkhan | 40 | | | |
| Hamidieh | 22 | | | |
| Chamgaz | 350 | Darreshahr | | 670 |
| Doab | 1310 | | | |
| Varayeneh | 1760 | Hamedan | | 1741 |
| | | Borujerd | | 1630 |
| Kheirabad | 1763 | Malayer | | 1778 |
| | | Eyvan | | 1200 |





|  |  | Kangavar | 1468 |
|---|---|---|---|
|  |  | Nahavand | 1680 |
| Ravansar | 1388 | Kamyaran | 1404 |
| Mahidasht | 1360 | Eslamabad | 1349 |
| Holeilan | 950 | Ilam | 1340 |
| Dartoot | 703 |  |  |
| Polchehr | 1280 | Kermanshah | 1306 |


**Table 4: Streamflow stations and characteristics**

| Sub-basin | Station | Long. ºE | Lat. ºN | Altit. (m) | Record length | Mean annual streamflow (mm/year) | Standard deviation (mm/year) |
|---|---|---|---|---|---|---|---|
| Qarasou | Ghoorbaghestan | 47.25 | 34.23 | 1300 | 1975–2011 | 111 | 49.0 |
| Gamasiab | Polchehr | 47.43 | 34.33 | 1306 | 1970–2011 | 81 | 38.9 |
| Kashkan | Poldokhtar | 47.72 | 33.17 | 650 | 1980–2011 | 158 | 63.2 |
| Seimareh | Nazarabad | 47.43 | 33.17 | 559 | 1979–2011 | 97 | 41.2 |
| Upper Karkheh | Payepol | 48.15 | 32.42 | 90 | 1974–2011 | 114 | 49.2 |

**Table 5: Average discharge variations from the pre-change period (Q1) to the post-change period (Q2)**
**and the magnitude of the change**

| Sub-basin | Break-point | Q1 (mm/year) | Q2 (mm/year) | ΔQ (mm/year) | ΔQ (%) |
|---|---|---|---|---|---|
| Qarasou | 1994 | 146 | 77.9 | - 68.1 | -47% |
| Gamasiab | 1995 | 107.8 | 57.7 | - 50.1 | -47% |
| Kashkan | 1993 | 186.2 | 138.5 | - 47.7 | -26% |
| Seimareh | 1995 | 124.2 | 72.3 | - 51.9 | -42% |
| Upper Karkheh | 1995 | 145.4 | 85.6 | - 59.8 | -41% |



**Table 6: Climate variation during pre- and post-change periods. Number 1 and 2 denote pre-change and post-change period, respectively**

| Sub-basin | $E_{01}$(mm) | $E_{02}$ (mm) | $\Delta E_0$ | P1 (mm) | P2 (mm) | $\Delta P$ |
|---|---|---|---|---|---|---|
| Qarasou | 2098 | 2206 | +5% | 473 | 381 | -19% |
| Gamasab | 2021 | 2277 | +13% | 494 | 432 | -13% |
| Kashkan | 2202 | 2597 | +18% | 480 | 474 | +1% |
| Seimareh | 2073 | 2240 | +8% | 446 | 382 | -14% |
| Upper Karkheh | 2138 | 2344 | +9% | 454 | 393 | -13% |


**Table 7: Suggested range for the calibration parameters of HBV**

| Characteristic parameters | Units | Description | Suggested domain | Calibrated values |
|---|---|---|---|---|
| athorn | [mm/day °C] | General correction factor for E0 | 0.15 – 0.3 | 0.15 - 0.25 |
| beta | | Drainage from soil factor | 1- 4 | 1 - 1.4 |
| cflux | [mm/day] | Maximum capillary flow | 0 - 2 | 0 |
| cfmax | [mm/°C, day] | Snow melt factor | 2 – 5 | 2 – 5 |
| dttm | [°C] | Threshold temperature | -2° - 2° | -2° - -1.5° |
| fc | [mm] | Field capacity | 100 - 1500 | 430 – 830 |
| k4 | [Unit/ day] | Recession coefficient | 0.001 – 0.1 | 0.015 - 0.05 |
| khq | [Unit/ day] | Recession coefficient | 0.005 – 0.5 | 0.1 - 0.25 |
| lp | | Limit for potential evaporation | <=1 | 0.94 – 1 |
| maxbaz | | # days in the transformation routine | 0 - 7 | 0 – 1 |
| pcorr | | Precipitation correction Factor | | 0.8 - 0.95 |
| perc | [mm/day] | Percolation capacity | 0.01 - 6 | 0.9 – 4.6 |
| rfcf | | Rainfall correction factor | 0.8 – 1.3 | 1.05 – 1.25 |
| sfcf | | Snowfall correction factor | 0.7 – 1.4 | 0.7 – 0.9 |
| tt | [°C] | Threshold temperature | -2° - 2° | 1.5° - 2° |

**Table 8: Evaluation index (R2) for the sub-basins.**

| Basin | Qarasou | Gamasab | Kashkan | Seimareh | Upper Karkheh |
|---|---|---|---|---|---|
| $R^2$ Calibration | 0.74 | 0.81 | 0.66 | 0.77 | 0.71 |
| $\delta$ (mm/year) | 29.4 | 1.07 | -28.2 | 5.3 | 16.7 |






**Table 9: Comparison between HBV model and Budyko method for estimation of streamflow changes in the Karkheh sub-basins**

| Basins | Breakpoint | $\varepsilon_P$ | $\varepsilon_{E0}$ | $\Delta Q$ (mm) | HBV | | Budyko | |
|---|---|---|---|---|---|---|---|---|
| | | | | | $\Delta Qc$ (mm/year) | $\Delta QH$ (mm/year) | $\Delta Qc$ (mm/year) | $\Delta QH$ (mm/year) |
| Qarasou | 1994 | 1.61 | -0.61 | 68.1 | 40.8 | 27.3 | 41.8 | 26.3 |
| Gamasab | 1995 | 1.81 | -0.81 | 50.1 | 29.7 | 20.4 | 27.7 | 22.4 |
| Kashkan | 1993 | 1.50 | -0.50 | 47.7 | 14.6 | 33.1 | 15.3 | 32.4 |
| Seimareh | 1995 | 1.65 | -0.65 | 51.9 | 31.3 | 20.6 | 29.6 | 22.3 |
| Upper Karkheh | 1995 | 1.60 | -0.60 | 59.8 | 31.0 | 28.8 | 32.2 | 27.6 |

**Table 10: Percentage of land use class change in five different sub-basins.**

| Sub-basin | Land use Type | Land use 1987 (%) | Land use 1995 (%) | Land use 2012 (%) |
|---|---|---|---|---|
| Seimareh | Irrigated | 1.37 | 4.49 | 6.88 |
| | Rainfed | 11.84 | 10.91 | 8.39 |
| | Range and pasture | 10.71 | 9.81 | 12.24 |
| | Forest | 4.77 | 3.42 | 1.08 |
| | Urban | 0.09 | 0.17 | 0.24 |
| Qarasou | Irrigated | 2.29 | 2.66 | 2.98 |
| | Rainfed | 5.02 | 5.55 | 4.92 |
| | Range and pasture | 4.87 | 4.07 | 4.61 |
| | Forest | 0.57 | 0.41 | 0.13 |
| | Urban | 0.11 | 0.15 | 0.19 |
| Gamasiab | Irrigated | 4.28 | 5.11 | 5.91 |
| | Rainfed | 7.62 | 8.09 | 13.00 |
| | Range and pasture | 14.16 | 11.41 | 6.73 |
| | Forest | 0.32 | 0.89 | 0.04 |
| | Urban | 0.46 | 1.35 | 1.12 |
| Kashkan | Irrigated | 1.12 | 2.82 | 6.80 |
| | Rainfed | 9.12 | 8.33 | 5.58 |
| | Range and pasture | 7.81 | 7.09 | 8.50 |





| | | | | |
|---|---|---|---|---|
| | Forest | 4.10 | 3.85 | 1.15 |
| | Urban | 0.05 | 0.11 | 0.23 |
| Upper Karkheh | Irrigated | 0.09 | 0.90 | 1.08 |
| | Rainfed | 2.09 | 1.59 | 0.85 |
| | Range and pasture | 6.34 | 6.40 | 7.15 |
| | Forest | 0.74 | 0.37 | 0.07 |
| | Urban | 0.05 | 0.05 | 0.12 |


**Table 11: Land use classes (total) percentage in different years**

| | Years | | |
|---|---|---|---|
| Land use class | 1987 | 1995 | 2012 |
| Irrigated (%) | 9.2 | 16 | 17.7 |
| Rainfed (%) | 35.7 | 34.5 | 35.3 |
| Range and pasture (%) | 43.9 | 38.8 | 42.3 |
| Forest (%) | 10.5 | 8.9 | 2.7 |
| Urban (%) | 0.8 | 1.8 | 2.1 |

**Table 12: Streamflow changes to 10% error in the Budyko parameters**

| Basins | $\varepsilon_P$ | $\varepsilon_{E0}$ | n | $\bar{Q}$ (mm/year) | $\bar{P}$ (mm/year) | $\overline{E_0}$ (mm/year) | $dQ_p\%$ | $dQ_{E0}\%$ | $dQ_n\%$ |
|---|---|---|---|---|---|---|---|---|---|
| Qarasou | 1.61 | -0.61 | 0.8 | 111 | 424 | 2155 | 4.3 | -0.32 | 2.3 |
| Gamasab | 1.81 | -0.81 | 0.99 | 81 | 461 | 2157 | 3.3 | -0.32 | 2.7 |
| Kashkan | 1.50 | -0.50 | 0.7 | 158 | 477 | 2437 | 5 | -0.32 | 1.3 |
| Seimareh | 1.65 | -0.65 | 0.83 | 97 | 412 | 2162 | 3.9 | -0.3 | 2.1 |
| Upper Karkheh | 1.60 | -0.60 | 0.77 | 114 | 422 | 2248 | 4.5 | -0.32 | 2 |





**Figures**



**Figure 1: Location of The KRB in Iran. ET stands for evapotranspiration.**




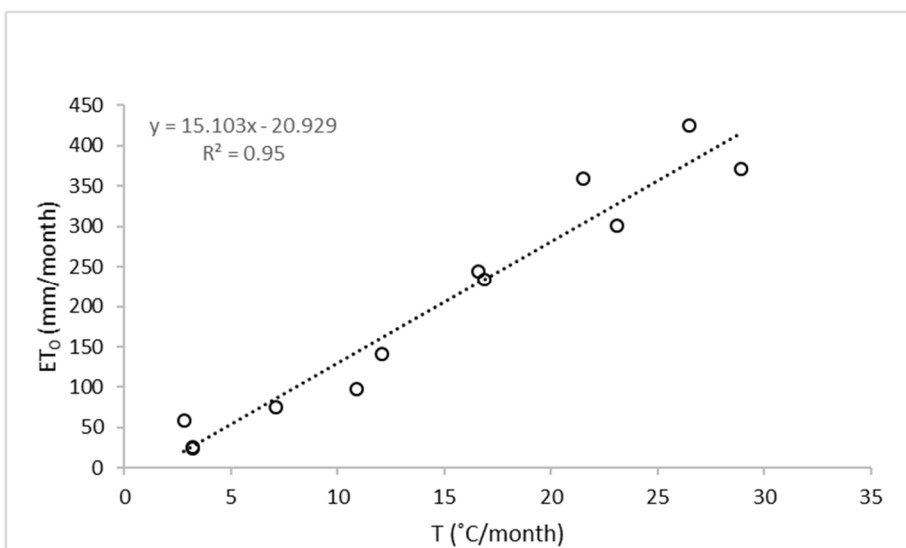

**Figure 2: R² between monthly evaporation (E0) and temperature (T) for the Polchehr reference station in 1995.**

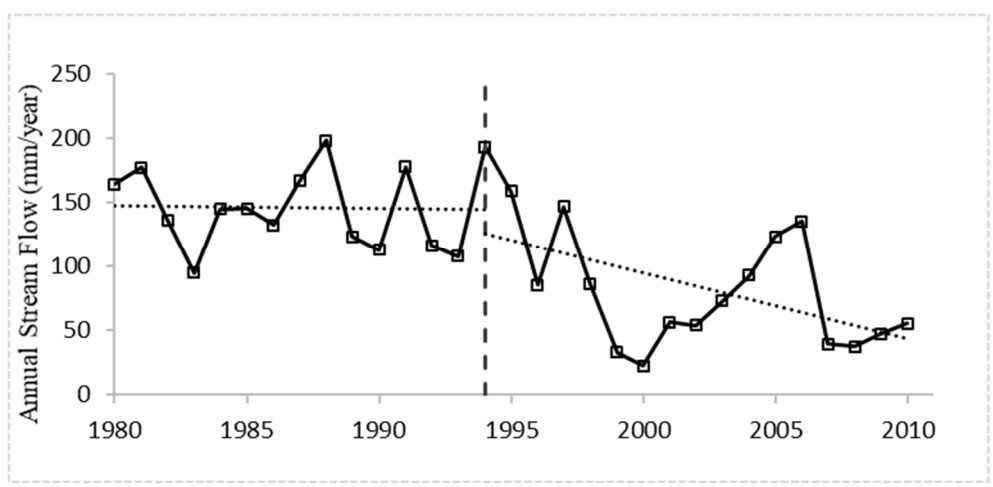

**Figure 3: Streamflow trend in the Qarasou sub-basin and the breakpoint detected by DBEST.**





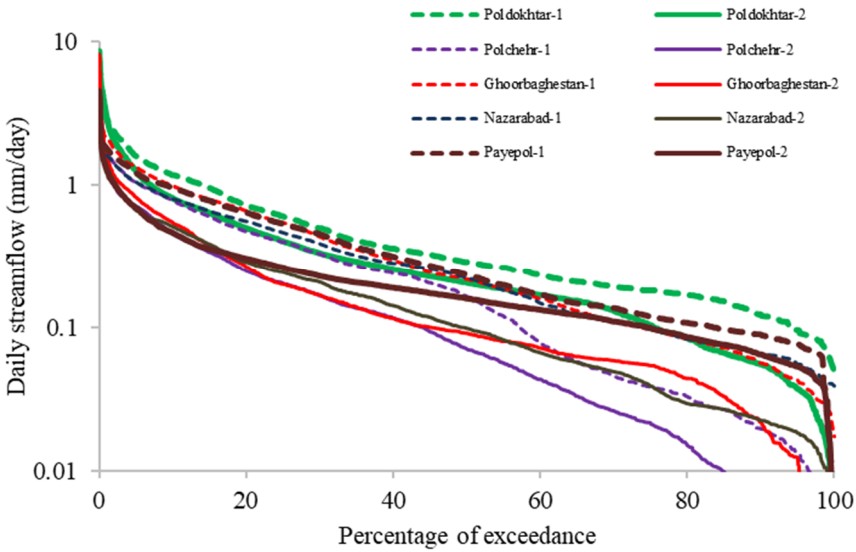

**Figure 4: FDCs of the Karkheh sub-basins for pre-change (1) and post-change (2) periods.**






**Figure 5: Mean monthly streamflow of the Karkheh sub-basins during pre-change (1) and post-change periods (2).**





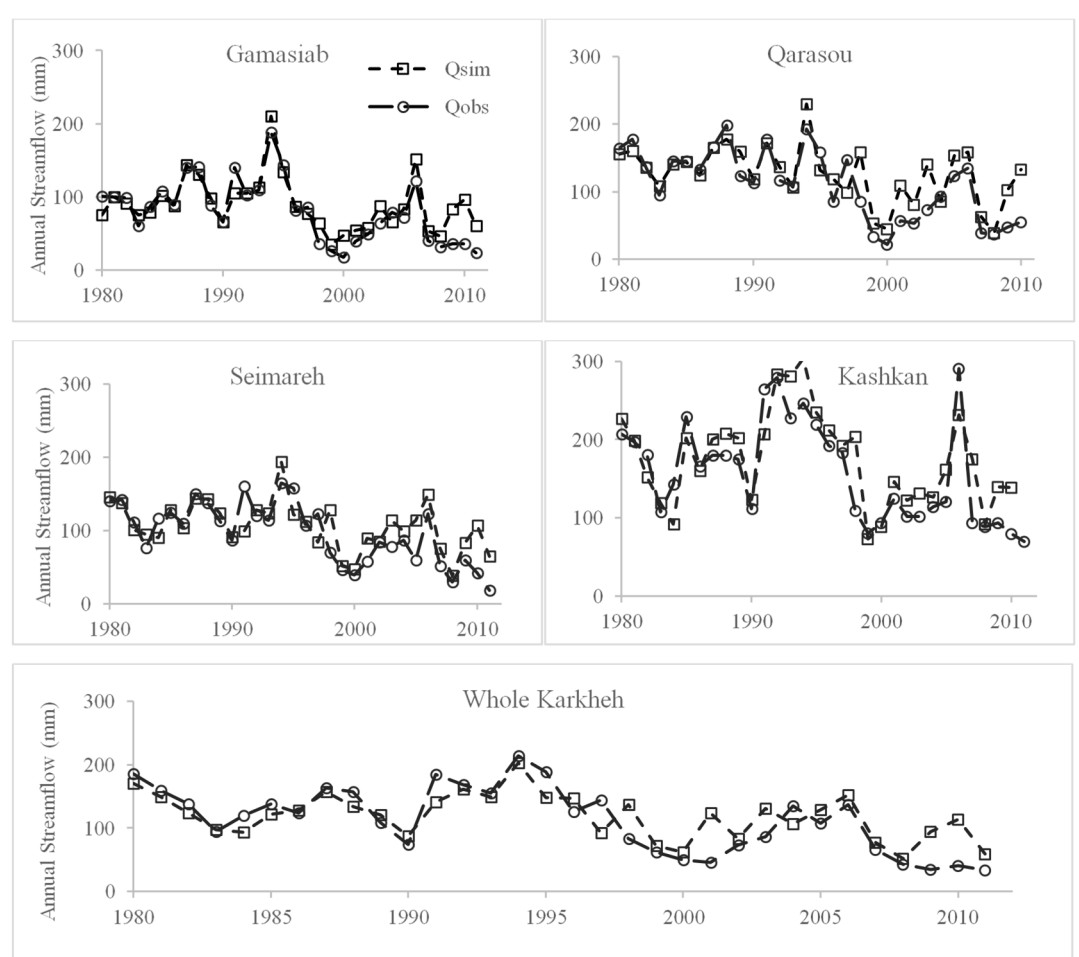


**Figure 6: Simulated vs. observed annual streamflow for the KRB.**



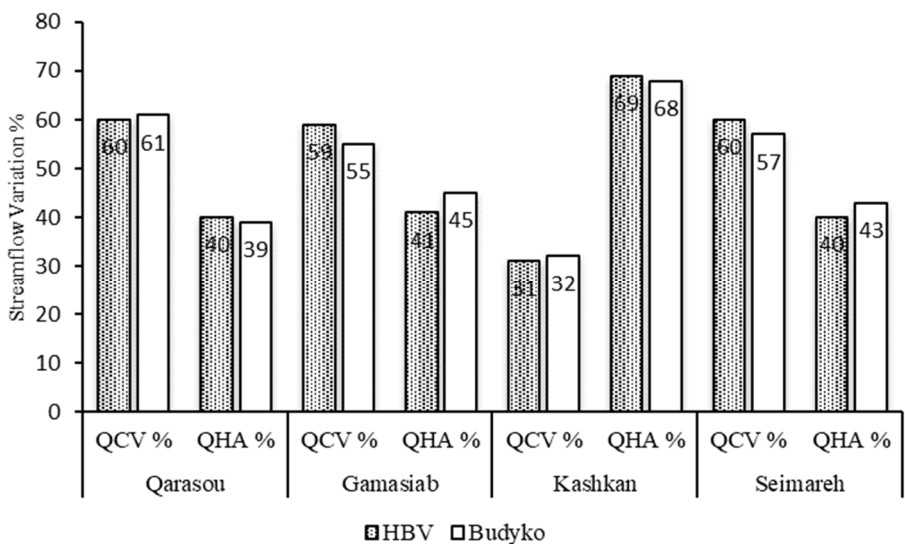

**Figure 7: Streamflow changes for all sub-basins due to climate variability and human activities. CV and HA denote climate change and human activities, respectively.**

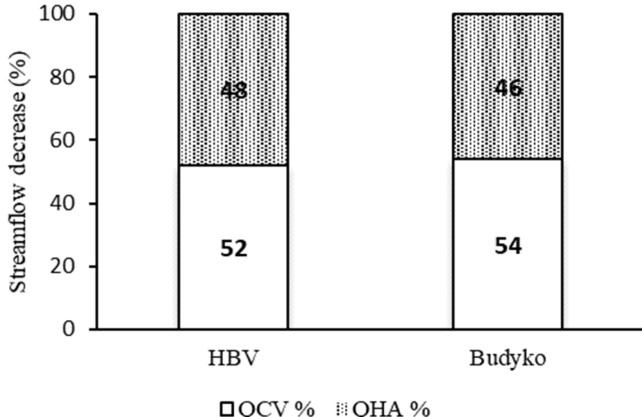

**Figure 8: Streamflow decrease (%) for the entire KRB due to climate variability and human impacts.**




**Figure 9: Figure 9: Land use maps for the KRB, (a) before breakpoint (1987), (b) breakpoint (1995), and (c) after breakpoint (2012).**



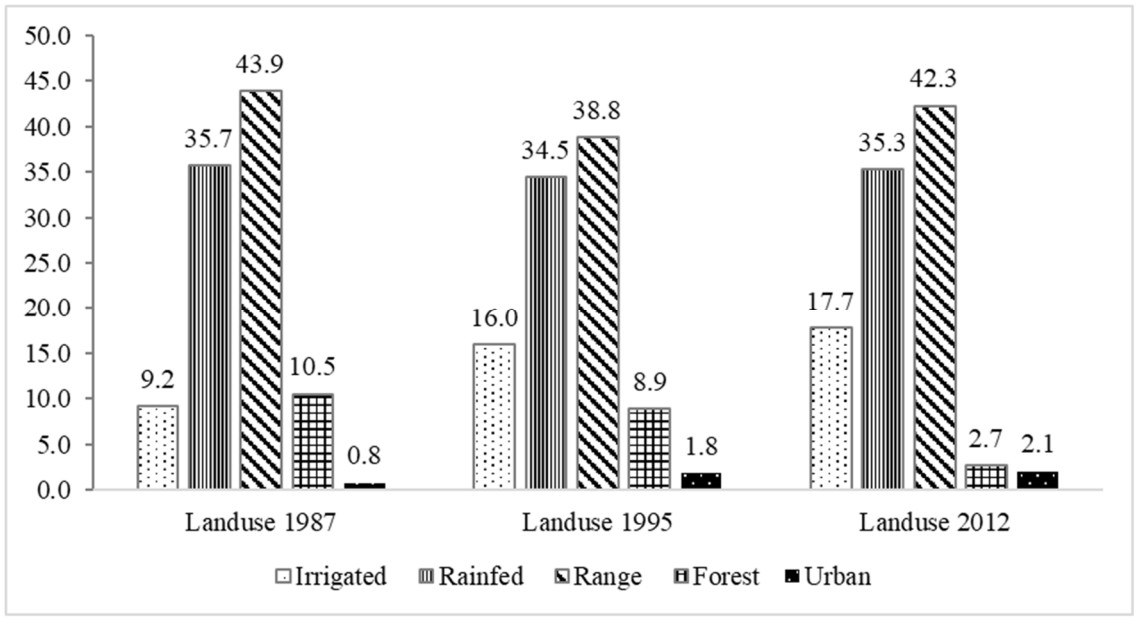

**Figure 10: From left to right, KRB land use classifications before breakpoint (1987), during the change (1995), and after breakpoint (2012).**

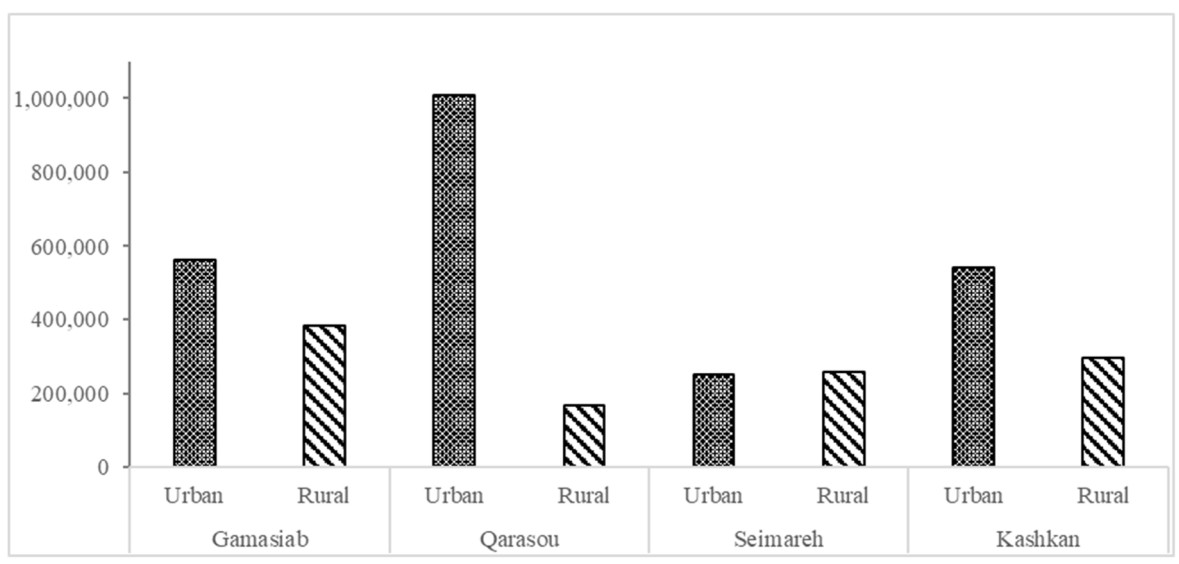

**Figure 11: Population distribution in the KRB according to the 2015 national record.**




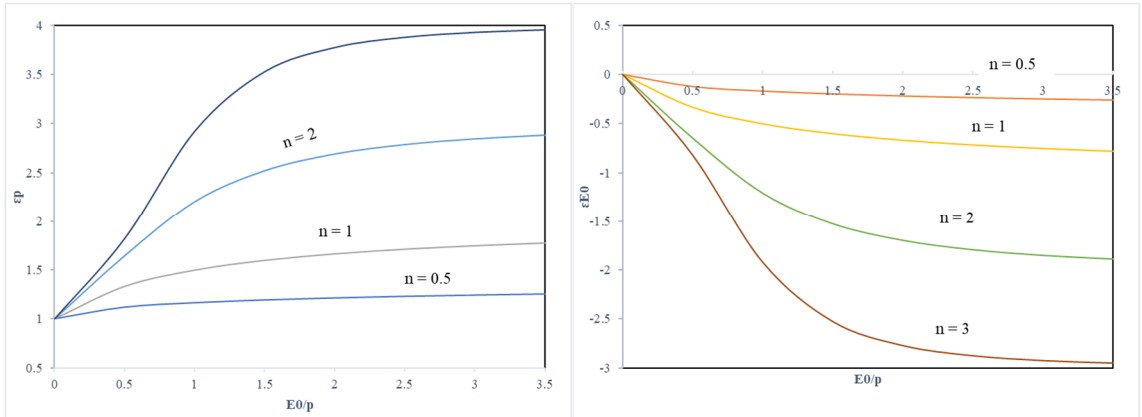

**Figure 12: Budyko equation curve responding to different values of n.**