# Peer review of "Assessment of streamflow decrease due to climate vs. human influence in a semiarid area"

_Hydrology and Earth System Sciences, 2019_

## Referee Comment (RC1) · Anonymous Referee #1 · 22 Feb 2020

Review of "**Assessment of streamflow decrease due to climate vs. human influence in a semiarid area**"

By Hamideh Kazemi, Hossein Hashemi, Fatemeh Fadia Maghsood, Seyyed Hasan Hosseini, Ranjan Sarukkalige, Sadegh Jamali, and Ronny Berndtsson

Thanks for the chance to review this paper.

**Message and contributions of the manuscript:** This study aims to estimate the fractional contribution of climate variation and regional human activities to changes in streamflow in a semi-arid river basin in Iran (Karkheh River Basin, KRB) over the period 1980-2012. In doing so, they have split the study period into two period of pre- and post-change based on streamflow trends. To estimate streamflow changes authors used a Budyko model together with the HBV hydrological model. They have also estimated land use changes in the basin showing that the irrigated farmlands expanded over this period. The study is done across the 5 sub-basins of KRB. This basin has regional importance due to energy and food production for over 4 million people.

**My evaluation:** I have not identified any major contributions in this study whether in terms of process understanding or methods. I acknowledge that authors said that this is the first time the Budyko hypothesis is used in Iran, which is a positive that helps to improve the geographical bias in the hydroclimatology literature. That said, all the methods (breakpoint detection, Budyko hypothesis, HBV modeling) have been developed previously. Similar studies on fractional contribution of climate variation and human activities to annual runoff changes have been done before, such as (Liu et al., 2017) and (Yang & Yang, 2011), each offering a new contribution either in method or process understanding. Even compared to those studies, the present study is an over simplification when it comes to analyses such as estimation of potential evapotranspiration (PET) based on a just a correlation with temperature, inadequate model calibration with no account for parameter uncertainty, and inadequate sensitivity analysis of the Budyko hypothesis (section 3.5). In addition to improving the analysis, the manuscript itself also requires major improvement: the paper is long with repetition of previous studies. The present discussion (section 4) is just an overview of the study area, a proper discussion of the results of this study is required. In doing so, the results should be discussed within the a broader context of studies in similar regions in Iran or around the globe. The discussion section would also benefit from a new sub-section discussing the limitations of results and suggestions for future studies. There are several typos, hence a careful proof read.

One major conceptual point is that the terms "climate change" and "climate variation" are used interchangeably in the manuscript. Regardless of what term is better suited (maybe climatic changes?), it is important to clearly define in the manuscript (perhaps in the introduction) that in this study climate change/variation – which human activities are compared to – is a lump variable that encompasses both *human-induced climate change* and *natural climatic variability*. Therefore, the contribution of human activities are under-estimated. This is an important point to discuss in the discussion section (see my comment 1 for details).

I am sorry that my evaluation is not as positive as authors would like. I genuinely acknowledge all the efforts that go into doing a research and writing a manuscript, yet in my opinion the current manuscript does not meet up to the requirement of HESS. This manuscript, upon improvement, better suits journals like *Journal of Hydrology: Regional Studies*.

**My detailed comments:**

1. Use a consistent term in referring to climatic changes throughout the manuscript (including title). And clearly define that what you referred to as climate change/variation is in fact a lump variable that includes both human-induced climate change and natural climatic variabilities. Therefore, there is a contribution of human activities to climatic changes that you are not capturing in this study, and hence human activities, in general, are under-estimated in this framework. What you estimated as "human activities", is in fact "regional human activities/interventions" in terms of regional water resources projects which is distinct from emissions that are also due to human activities. Other factors such as the impact of dam construction on regional evaporation and precipitation is assumed to be zero, while a major dam as such would presumably have impacts on the regional climate. Human activities are also under-estimated within the Budyko model by definition, as the parameter n in the Budyko equation is assumed to be a constant. That is, interannual changes in catchment characteristics are assumed to be zero (Yang & Yang, 2011). That means the pre-change period could have been impacted by regional human activities that are hard to capture and compare with the post-change period within the present method.

   a. On this topic, there been several recent studies on Lake Urmia, not far from KRB (Alizade Govarchin Ghale et al., 2018; Alizade Govarchin Ghale et al., 2019; Chaudhari et al., 2018; Fazel et al., 2017; Khazaei et al., 2019). Fazel et al. (2017) reported an interesting point, relevant to your study, that "*flow regime in river headwaters appeared to be dominated by natural forces, … the reduction in river flow magnitude increased from headwaters to downstream reaches for all rivers*". This is a major point to bear in mind when discussing your results, as you have not differentiated between headwater and downstream changes in flow.

2. The introduction requires further refinement to be sharp and concise.

3. There is no shortage of method for estimating potential evapotranspiration, including plenty of temperature-based methods (McMahon et al., 2013). A simple correlation is an over-simplification. The T-PET is not a linear relationship. Check out the modified Hargreaves method Droogers and Allen (2002) or the temperature-based method in Oudin et al. (2005).

4. I'd take issue with your hydrological modelling approach on two main grounds. (1) Parameter uncertainty is not accounted for. It is well-established that hydrological models require some type of parameter uncertainty analysis, i.e. models should be used as ensembles. In addition, it is not clear whether you have calibrated the model on daily scale or monthly. Calibrating the model on monthly scale would introduce additional uncertainties. It is easier to achieve a higher model performance in monthly calibrations, as daily variations would be ignored. This is particularly important when models are transferred to changing conditions, as in this study. A model that

cannot reproduce daily variations in the calibration period, would do even worse during an evaluation period with considerable changed conditions. (2) The objective functions used to evaluate the model performance are inadequate. $R^2$, which is essentially NSE (Nash Sutcliffe efficiency), has been demonstrated to be an inadequate metric for evaluating model performance (Gupta et al., 2009; Murphy, 1988). KGE (Kling Gupta Efficiency, updated version by Kling et al. (2012)) has been shown to be a better alternative to NSE, which already includes a bias term. So, if KGE is used, the second metric you used (Equation 3) would be redundant and you can remove it from model calibration/evaluation. So, I'd suggest to redo the hydrological modeling to account for parameter uncertainty using the KGE metric – and on a daily basis.

   a. Details about HBV model particularly the name of the parameters are unnecessary. Simply remove them and refer to original literature of "the version" of the model you used in this study (there are different versions of HBV). Other parts such as Table 7 (range of calibrated parameters) should be in the supplements as they are not contributing to the main story of the paper.

   b. Lines 356-360 "*The HBV model overestimated the streamflow for the post-change period, which suggests that there existed factors other than climate variations affecting the streamflow of the study basins. These factors are believed to be related to human activities*". You are oversimplifying the problem here. The issue that hydrological models perform poorly under changing conditions is a widely recognized issue and in fact one of the unsolved question of hydrological sciences (Question 19 in Blöschl et al., 2019). That said, you cannot simply conclude that those changes are due to human activities. The underlying reason is a combination of model structure adequacy to represent the evolution of catchment processes, model parameterization, climatic changes (whether natural variability or human-induced) that cannot be represented well by model structure and parameters, and changes due to human activities. I agree that human activities has a role in this, but you cannot simply "believe" that it is all due to human activities.

5. Section 2.5 on Budyko method is too long. All the equations have been derived previously. You can simply explain the method conceptually, direct readers to the literature for more details, and present only the equations you used.

   a. Line 241, it is not "believed" but "assumed" to be…

6. Section 2.7, lines 288-289, "*In order to adequately simulate a hydrological response at the basin level, accurate data such as climate variables (precipitation, ET, etc.) and catchment physical characteristics (topography, land coverage, vegetation, etc.) are vital.*" what about the adequacy of the model structure, both the Budyko model and HBV, in representing catchment processes and change?

   a. Lines 293-294, it does not make sense and does not follow the previous paragraph. Climate elasticity does not address the uncertainties…?

   b. The possible sources of errors (line 300) are the break change method, Budyko model and parameterization, HBV model structure and parameterization, and data

quality and length. Each of these factors, individually or combined, may change the results to some degree. Among these factors, you did not look into the dependency of your results upon the choice of breakpoint method. For instance Liu et al. (2017) used 8 different methods to account for this. You have not looked into HBV parameter uncertainty. What you discussed in section 3.5 is actually a sensitivity analysis of the Budyko model, for which you only reported 10% change in parameter n. What if all the inputs and parameters of Budyko are subjected to 5, 10 and 15% change? How the change in Budyko inputs propagate to the final answer? What about the change in model structure, e.g. if another model was used instead of HBV? Etcetera etcetera… A more rigorous sensitivity analysis is required.

7. The discussion of land use changes (section 3.4) can be improved. It takes one page, two figures and two tables to just say that irrigation area has increased significantly while forested area decreased. It is also good to briefly mention that there are three snapshots of an early year, changepoint, and final year to help readers follow the story line.

8. The figures are not properly ordered. After presenting figure 2 on page 5, authors jumped to figure 9 on page 10, and then back to figure 3 (page 10). Figure 11 is not relevant to the discussion session of the paper, it is relevant to introduction or study area. Figure 12 does not offer anything new, Yang and Yang (2011) already presented and discussed this.
   a. Figure 3, present all the 5 sub-basins in one figure.
   b. Figure 6 is not easy to read.
   c. Figure 7 and 8 could be combined, as two subplots.

9. Line 469, "*The Budyko method showed to be a reliable … method to analyze streamflow changes during the study period*". You have not offered any analysis nor discussion on the *reliability* of the Budyko method.

10. There are several typos throughout the text. A carful proof read is required. Here are a few minor corrections and typos (just as examples):
    a. The name of "Seimareh" river has two different spellings on the Figure and in the text.
    b. The symbol for changes due to human activities is not consistent: $\Delta Q_{HA}$ (e.g. line 243) and $\Delta Q_H$ (e.g. equation 11) – fix throughout. Also fix the symbol for $ET_0$ and $E_0$ in equation 13. And the denominator in the second term of the equation 15.
    c. Avoid making up abbreviations in the middle of the manuscript such as *Eqn* on line 202. Be consistent either use abbreviation or the full word throughout. Check the journal guideline for this. Same comment applies to Fig. and Figure.

11. Minor comments:
    a. Within the current visualization of Figure 1, it is very hard to see the stream gauges in most sub-basins. Since you didn't mark each discharge station with its name on the map, it is hard for readers to tell where Payepol station is: is it the blue square inside the Upper Karkheh sub-basin, or is it the one outside of it (the final outlet of the entire catchment)? Also show the location of the Karkheh dam on the figure.

b. Study area page 3: Line 93, I think it's better to mention the actual population (i.e. 4+ million people) the river supplies water to. It's good to keep "5% of Iran's population" for context too. Line 103: it is good to point out that more than ~60% of the precipitation received in northern mountainous area is snow.

**References**

Alizade Govarchin Ghale, Y., Altunkaynak, A., & Unal, A. (2018). Investigation Anthropogenic Impacts and Climate Factors on Drying up of Urmia Lake using Water Budget and Drought Analysis. *Water Resources Management, 32*(1), 325-337. doi:10.1007/s11269-017-1812-5

Alizade Govarchin Ghale, Y., Baykara, M., & Unal, A. (2019). Investigating the interaction between agricultural lands and Urmia Lake ecosystem using remote sensing techniques and hydro-climatic data analysis. *Agricultural Water Management, 221*, 566-579. doi:https://doi.org/10.1016/j.agwat.2019.05.028

Blöschl, G., Bierkens, M. F. P., Chambel, A., Cudennec, C., Destouni, G., Fiori, A., et al. (2019). Twenty-three Unsolved Problems in Hydrology (UPH) – a community perspective. *Hydrological Sciences Journal*. doi:https://doi.org/10.1080/02626667.2019.1620507

Chaudhari, S., Felfelani, F., Shin, S., & Pokhrel, Y. (2018). Climate and anthropogenic contributions to the desiccation of the second largest saline lake in the twentieth century. *Journal of Hydrology, 560*, 342-353. doi:https://doi.org/10.1016/j.jhydrol.2018.03.034

Droogers, P., & Allen, R. G. (2002). Estimating Reference Evapotranspiration Under Inaccurate Data Conditions. *Irrigation and Drainage Systems, 16*(1), 33-45. doi:10.1023/a:1015508322413

Fazel, N., Torabi Haghighi, A., & Kløve, B. (2017). Analysis of land use and climate change impacts by comparing river flow records for headwaters and lowland reaches. *Global and Planetary Change, 158*, 47-56. doi:https://doi.org/10.1016/j.gloplacha.2017.09.014

Gupta, H. V., Kling, H., Yilmaz, K. K., & Martinez, G. F. (2009). Decomposition of the mean squared error and NSE performance criteria: Implications for improving hydrological modelling. *Journal of Hydrology, 377*(1–2), 80-91. doi:http://dx.doi.org/10.1016/j.jhydrol.2009.08.003

Khazaei, B., Khatami, S., Alemohammad, S. H., Rashidi, L., Wu, C., Madani, K., et al. (2019). Climatic or regionally induced by humans? Tracing hydro-climatic and land-use changes to better understand the Lake Urmia tragedy. *Journal of Hydrology, 569*, 203-217. doi:https://doi.org/10.1016/j.jhydrol.2018.12.004

Kling, H., Fuchs, M., & Paulin, M. (2012). Runoff conditions in the upper Danube basin under an ensemble of climate change scenarios. *Journal of Hydrology, 424-425*, 264-277. doi:https://doi.org/10.1016/j.jhydrol.2012.01.011

Liu, J., Zhang, Q., Singh, V. P., & Shi, P. (2017). Contribution of multiple climatic variables and human activities to streamflow changes across China. *Journal of Hydrology, 545*, 145-162. doi:https://doi.org/10.1016/j.jhydrol.2016.12.016

McMahon, T. A., Peel, M. C., Lowe, L., Srikanthan, R., & McVicar, T. R. (2013). Estimating actual, potential, reference crop and pan evaporation using standard meteorological data: a pragmatic synthesis. *Hydrol. Earth Syst. Sci., 17*(4), 1331-1363. doi:10.5194/hess-17-1331-2013

Murphy, A. H. (1988). Skill Scores Based on the Mean Square Error and Their Relationships to the Correlation Coefficient. *Monthly Weather Review, 116*(12), 2417-2424. doi:10.1175/1520-0493(1988)116<2417:ssbotm>2.0.co;2

Oudin, L., Hervieu, F., Michel, C., Perrin, C., Andréassian, V., Anctil, F., et al. (2005). Which potential evapotranspiration input for a lumped rainfall–runoff model?: Part 2—Towards a simple and efficient potential evapotranspiration model for rainfall–runoff modelling. *Journal of Hydrology, 303*(1–4), 290-306. doi:http://dx.doi.org/10.1016/j.jhydrol.2004.08.026

Yang, H., & Yang, D. (2011). Derivation of climate elasticity of runoff to assess the effects of climate change on annual runoff. *Water Resources Research, 47*(7). doi:10.1029/2010wr009287

---

## Referee Comment (RC2) · Anonymous Referee #2 · 24 Mar 2020

The manuscript is about using Budyko method and HBV model to investigate mean annual streamflow changes, due to climate variation and human influence, in the important Karkheh River Basin in western Iran. Although this manuscript is an interesting study but this study doesn't identify any major contributions in terms of process understanding or developing new methods. However as stated in the manuscript, authors claim that their approach combining HBV and Budyko is novel and for the first time used in Iran. However, the knowledge gap/novelty and the importance of this work is still missing throughout the whole manuscript and it needs to be clearly stated. The method section is too long with elaborated with details. In other hand, the discussion (section is too short and general repeating the same message said earlier rather than

putting the results from this study in a broader context of studies in similar regions and worldwide.

In the conclusion, authors claim that "The outcome of this study can be used to assist policymakers and water professionals in proposing a proper water management plan to prevent the further reduction of streamflow and groundwater storage". How the results of this study would help policy makers to prevent reduction of streamflow and groundwater storage? When the results show that we have a combined effect of both human (increased irrigated area and reduction of forests), and climate (decreasing annual precipitation) on streamflow reduction almost all over the basins (20).

Authors need to work properly with all figures for instance order of figures should be improved, authors refer to figure 2 and then figure 9 and then back to figure 3. Figure 11 can be removed from the discussion part and Figure 12 is not necessary.

This a big assumption in this work that streamflow has not been influenced by human activities before the breakpoint. Please clarify!

I believe that this manuscript does not warrant a publication in HESS now and requires intensive additional work to be modified and re-considered for any possible publication in future. .

---

## Author Comment (AC1) · 29 Apr 2020

**Response to the reviewer's comments - hess-2019-618**

The authors would like to thank the reviewers for their constructive comments that helped to improve the quality of the manuscript. Our point-by-point responses for the reviewers' valuable comments are listed below.

**Reviewer 1:**

*I have not identified any major contributions in this study whether in terms of process understanding or methods. I acknowledge that authors said that this is the first time the Budyko hypothesis is used in Iran, which is a positive that helps to improve the geographical bias in the hydroclimatology literature. That said, all the methods (breakpoint detection, Budyko hypothesis, HBV modeling) have been developed previously. Similar studies on fractional contribution of climate variation and human activities to annual runoff changes have been done before, such as (Liu et al., 2017) and (Yang & Yang, 2011), each offering a new contribution either in method or process understanding.*

Thank you for the detailed review and pinpointing shortcoming of the manuscript.

Although, we admit that all the methods including the Budyko, HBV, DBEST, and satellite image processing used in this study are separately developed in previous studies, our manuscript introduces a novel combination of these methods such that a new, more robust framework of separating human vs climate variation effects on streamflow of large river basins in the data-scarce area is presented.

As mentioned, this is the first time that Budyko analysis is implemented in the Iranian catchment. The case study is one of the most important catchments in the country and ironically there is the lack of studies investigating separate impacts of human activities and climate change in the area. In any case, the study offers a new approach for data scarce areas to quantify effects of climate change versus anthropogenic influence. This involves a technique to validate the Budyko method with remote sensing analyses. None of the previous studies mentioned by the reviewer has applied satellite remote sensing techniques to quantify land use changes over a long period of time for verification of the Budyko and HBV modelling results. We firmly believe that this is one of the major contributions of this study to the methodological approaches in the field of hydrology.

Also, as a novel approach, we used the newly developed DBEST algorithm for detecting breakpoints. DBEST uses a (novel) segmentation algorithm for detecting and characterizing significant breakpoints, and has a general designation making it suitable for different time series data inputs. It has been applied successfully in several other studies and this paper shows its suitability for streamflow change detection. Not mentioning that the introduction of the DBEST method here is important because it is new in hydrological studies and specifically relevant in Budyko applications for a more reasonable, systematic selection of time periods to be compared.

Even compared to those studies, the present study is an over simplification when it comes to analyses such as estimation of potential evapotranspiration (PET) based on a just a correlation with temperature, inadequate model calibration with no account for parameter uncertainty, and inadequate sensitivity analysis of the Budyko hypothesis (section 3.5).

The sensitivity analysis provided in section 3.5 and Table 11 shows that the model is not very sensitive to changes in evapotranspiration and hence temperature. Moreover, the new PET stations were added to increase the accuracy of the evapotranspiration time series. As the reviewer pointed out, we used a traditional, however, scientifically sound method for estimating ET at the basin level, even though this was not the main focus of the study.

In addition to improving the analysis, the manuscript itself also requires major improvement: the paper is long with repetition of previous studies.

Thanks for the comment, we have done an intensive revision; we have shortened the manuscript and removed the repetitive section highlighted in the revised version of the manuscript.

The present discussion (section 4) is just an overview of the study area, a proper discussion of the results of this study is required. In doing so, the results should be discussed within the a broader context of studies in similar regions in Iran or around the globe. The discussion section would also benefit from a new sub-section discussing the limitations of results and suggestions for future studies. There are several typos, hence a careful proof read.

The authors agree and the discussion section is improved in the revised version (page 12-14 – lines 345-445). Further, a sub-section was added to the discussion part, on page 14- lines 410-415, improving the analyses of the results.

One major conceptual point is that the terms "climate change" and "climate variation" are used interchangeably in the manuscript. Regardless of what term is better suited (maybe climatic changes?), it is important to clearly define in the manuscript (perhaps in the introduction) that in this study climate change/variation – which human activities are compared to – is a lump variable that encompasses both human-induced climate change and natural climatic variability. Therefore, the contribution of human activities are under-estimated. This is an important point to discuss in the discussion section (see my comment 1 for details). Use a consistent term in referring to climatic changes throughout the manuscript (including title).

And clearly define that what you referred to as climate change/variation is in fact a lump variable that includes both human-induced climate change and natural climatic variabilities. Therefore, there is a contribution of human activities to climatic changes that you are not capturing in this study, and hence human activities, in general, are under-estimated in this framework. What you estimated as "human

activities", is in fact "regional human activities/interventions" in terms of regional water resources projects which is distinct from emissions that are also due to human activities. Other factors such as the impact of dam construction on regional evaporation and precipitation is assumed to be zero, while a major dam as such would presumably have impacts on the regional climate.

Although it is not uncommon to use the terms, climate change and climate variability, interchangeably, but, the authors preferred to use the climate variability, because the study period is not long, in climate scale.

Definitely, human activities and climate change are tied together and not independent. However, to do the analysis, it is a common assumption to consider them as independent parameters (Zeng, Xia et al. 2014, Zhang, Liu et al. 2016, Wu, Miao et al. 2017).

In this study, as you mentioned, climate variability is a combination of variability of climate, as well as global-scale human-induced effects due to worldwide greenhouse gasses increase. Human activities, on the other hand, is regional impacts of human. Dam construction and agricultural activities, etc. are intimately embedded in human impact.

Many of these relationships, now have been better described and discussed in the revised manuscript.

Human activities are also under-estimated within the Budyko model by definition, as the parameter n in the Budyko equation is assumed to be a constant. That is, interannual changes in catchment characteristics are assumed to be zero (Yang & Yang, 2011).

Similar studies have assuemed the parameter n as a constant. We have now discussed this limitation on page 15- line 425.

That means the pre-change period could have been impacted by regional human activities that are hard to capture and compare with the post-change period within the present method. a. On this topic, there been several recent studies on Lake Urmia, not far from KRB (Alizade Govarchin Ghale et al., 2018; Alizade Govarchin Ghale et al., 2019; Chaudhari et al., 2018; Fazel et al., 2017; Khazaei et al., 2019). Fazel et al. (2017) reported an interesting point, relevant to your study, that "flow regime in river headwaters appeared to be dominated by natural forces, … the reduction in river flow magnitude increased from headwaters to downstream reaches for all rivers". This is a major point to bear in mind when discussing your results, as you have not differentiated between headwater and downstream changes in flow. 2.

Definitely upper flow and downstream play important roles in a catchment's flow regime and we have considered them too, in this work. The basin was divided into five sub-basins and the upper basin streamflow was taken into account while doing analysis for the lower sub-basin.

The introduction requires further refinement to be sharp and concise. 3. There is no shortage of method for estimating potential evapotranspiration, including plenty of temperature-based methods (McMahon

et al., 2013). A simple correlation is an oversimplification. The T-PET is not a linear relationship. Check out the modified Hargreaves method Droogers and Allen (2002) or the temperature-based method in Oudin et al. (2005). 4.

We have improved the introduction. As mentioned, our results are not that sensitive to changes in evaporation. Moreover, we had several evapotranspiration stations in the catchment. The extra stations are added to improve the accuracy of the estimations.

I'd take issue with your hydrological modelling approach on two main grounds. (1) Parameter uncertainty is not accounted for. It is well-established that hydrological models require some type of parameter uncertainty analysis, i.e. models should be used as ensembles.

We have now changed the previous HBV modelling results and applied the most recent version of HBV model (HBV-light) which has the facility to integrate the "uncertainty analysis". Using the embedded Genetic Algorithm in the HBV-light model, we were able to perform an uncertainty analysis using the 10 best parameter sets. The results of the investigated uncertainty is added in the revised version of the manuscript (subsection on page 12).

In addition, it is not clear whether you have calibrated the model on daily scale or monthly. Calibrating the model on monthly scale would introduce additional uncertainties. It is easier to achieve a higher model performance in monthly calibrations, as daily variations would be ignored. This is particularly important when models are transferred to changing conditions, as in this study. A model that cannot reproduce daily variations in the calibration period, would do even worse during an evaluation period with considerable changed conditions.

The HBV model simulation including calibration process and all inputs are on a daily basis. In the figures, they are shown on a monthly time-scale for the sake of presentation. We have now discussed this for clarification on page 9 line 279.

(2) The objective functions used to evaluate the model performance are inadequate. R2, which is essentially NSE (Nash Sutcliffe efficiency), has been demonstrated to be an inadequate metric for evaluating model performance (Gupta et al., 2009; Murphy, 1988). KGE (Kling Gupta Efficiency, updated version by Kling et al. (2012)) has been shown to be a better alternative to NSE, which already includes a bias term. So, if KGE is used, the second metric you used (Equation 3) would be redundant and you can remove it from model calibration/evaluation. So, I'd suggest to redo the hydrological modeling to account for parameter uncertainty using the KGE metric – and on a daily basis.

Thank you for the suggestion. We did the hydrological modelling again with the new version of HBV model and added the mentioned criteria (KGE) in Table 8, based on your suggestion.

a. Details about HBV model particularly the name of the parameters are unnecessary. Simply remove them and refer to original literature of "the version" of the model you used in this study (there are different versions of HBV). Other parts such as Table 7 (range of calibrated parameters) should be in the supplements as they are not contributing to the main story of the paper.

Thanks for the comment. We have now revised the manuscript accordingly. The introduction is enhanced and more specific now. Unnecessary and wordy parts have been removed and the table has been moved to Appendix section on page 33.

Lines 356-360 "The HBV model overestimated the streamflow for the post-change period, which suggests that there existed factors other than climate variations affecting the streamflow of the study basins. These factors are believed to be related to human activities". You are oversimplifying the problem here. The issue that hydrological models perform poorly under changing conditions is a widely recognized issue and in fact one of the unsolved question of hydrological sciences (Question 19 in Blöschl et al., 2019). That said, you cannot simply conclude that those changes are due to human activities. The underlying reason is a combination of model structure adequacy to represent the evolution of catchment processes, model parameterization, climatic changes (whether natural variability or human-induced) that cannot be represented well by model structure and parameters, and changes due to human activities. I agree that human activities has a role in this, but you cannot simply "believe" that it is all due to human activities.

Definitely, different researchers have different methods and understanding of the problem and concept. However, as mentioned by Dey et al. (2017), the applied procedure is a common method which has been employed by other researchers such as (Hu, Liu et al. 2012, Sun, Tian et al. 2014, Chang, Zhang et al. 2016). However, your valuable point has now been discussed in the limitation section in the revised manuscript.

5. Section 2.5 on Budyko method is too long. All the equations have been derived previously. You can simply explain the method conceptually, direct readers to the literature for more details, and present only the equations you used. a. Line 241, it is not "believed" but "assumed" to be…

Thanks for the comment. We revised the manuscript accordingly. 'Believed' was also changed to 'assumed'.

6. Section 2.7, lines 288-289, "In order to adequately simulate a hydrological response at the basin level, accurate data such as climate variables (precipitation, ET, etc.) and catchment physical characteristics (topography, land coverage, vegetation, etc.) are vital." what about the adequacy of the model structure, both the Budyko model and HBV, in representing catchment processes and change? a. Lines 293-294, it does not make sense and does not follow the previous paragraph.

Thanks for the comment. We reviewed and changed these lines and highlighted them in the text, uncertainty analysis subsection.

The possible sources of errors (line 300) are the break change method, Budyko model and parameterization, HBV model structure and parameterization, and data quality and length. Each of these factors, individually or combined, may change the results to some degree. Among these factors, you did not look into the dependency of your results upon the choice of breakpoint method. For instance Liu et al. (2017) used 8 different methods to account for this.

Definitely, each hydrological model has its own advantages and limitations and Budyko and HBV models have no exceptions.

Liu et al. (2017) studied several catchments across china, using three approaches of AMOC, Ordered Clustering Test, and Pettitt test methods for the detection of change points.

The focus of studies such as Liu et al. (2017) is to analyse, in detail, the driving factors and their exact shares affecting streamflow changes. While, in this study we basically focus on the impact of human activities and climate variation. For future studies, if there is enough data available, the details of human activities can be studied.

The breaking point occurs when the trend of the streamflow changes. In this study, the breakpoints were detected by a newly developed DBEST method and was cross validated with Man-Kendall test.

What you discussed in section 3.5 is actually a sensitivity analysis of the Budyko model, for which you only reported 10% change in parameter n. What if all the inputs and parameters of Budyko are subjected to 5, 10 and 15% change? How the change in Budyko inputs propagate to the final answer? What about the change in model structure, e.g. if another model was used instead of HBV?

The suggestion for future work and discussion has been added to the discussion part but implementing all mentioned methods are beyond the scope of this research.

The discussion of land use changes (section 3.4) can be improved. It takes one page, two figures and two tables to just say that irrigation area has increased significantly while forested area decreased. It is also good to briefly mention that there are three snapshots of an early year, changepoint, and final year to help readers follow the story line.

Thank you for your suggestion. It has now been added to the revised manuscript line 312.

The figures are not properly ordered. After presenting figure 2 on page 5, authors jumped to figure 9 on page 10, and then back to figure 3 (page 10). Figure 11 is not relevant to the discussion session of the paper, it is relevant to introduction or study area. Figure 12 does not offer anything new, Yang and Yang (2011) already presented and discussed this.

a. Figure 3, present all the 5 sub-basins in one figure.

b. Figure 6 is not easy to read.

c. Figure 7 and 8 could be combined, as two subplots.

Thanks for the precise comments. We amended the text and figures accordingly.

9. Line 469, "The Budyko method showed to be a reliable … method to analyze streamflow changes during the study period". You have not offered any analysis nor discussion on the reliability of the Budyko method.

There are several typos throughout the text. A carful proof read is required. Here are a few minor corrections and typos (just as examples):

a. The name of "Seimareh" river has two different spellings on the Figure and in the text.

b. The symbol for changes due to human activities is not consistent: $\Delta QHA$ (e.g. line 243) and $\Delta QH$ (e.g. equation 11) – fix throughout. Also fix the symbol for ET0 and E0 in equation 13. And the denominator in the second term of the equation 15.

c. Avoid making up abbreviations in the middle of the manuscript such as Eqn on line 202. Be consistent either use abbreviation or the full word throughout. Check the journal guideline for this. Same comment applies to Fig. and Figure.

We appreciate your careful reading of the manuscript. These comments are now implemented and highlighted in the revised version of the manuscript.

a. Within the current visualization of Figure 1, it is very hard to see the stream gauges in most sub-basins. Since you didn't mark each discharge station with its name on the map, it is hard for readers to tell where Payepol station is: is it the blue square inside the Upper Karkheh sub-basin, or is it the one outside of it (the final outlet the entire catchment)? Also show the location of the Karkheh dam on the figure.

Thanks for pinpointing the issues. A better map has now been used.

**Additional References (added to the revised manuscript):**

Chang, J., H. Zhang, Y. Wang and Y. Zhu (2016). "Assessing the impact of climate variability and human activities on streamflow variation." Hydrology and Earth System Sciences **20**(4): 1547-1560.
Hu, S., C. Liu, H. Zheng, Z. Wang and J. Yu (2012). "Assessing the impacts of climate variability and human activities on streamflow in the water source area of Baiyangdian Lake." Journal of Geographical Sciences **22**(5): 895-905.
Sun, Y., F. Tian, L. Yang and H. Hu (2014). "Exploring the spatial variability of contributions from climate variation and change in catchment properties to streamflow decrease in a mesoscale basin by three different methods." Journal of Hydrology **508**: 170-180.

Wu, J., C. Miao, X. Zhang, T. Yang and Q. Duan (2017). "Detecting the quantitative hydrological response to changes in climate and human activities." Science of the Total Environment **586**: 328-337.

Zeng, S., J. Xia and H. Du (2014). "Separating the effects of climate change and human activities on runoff over different time scales in the Zhang River basin." Stochastic environmental research and risk assessment **28**(2): 401-413.

Zhang, Q., J. Liu, V. P. Singh, X. Gu and X. Chen (2016). "Evaluation of impacts of climate change and human activities on streamflow in the Poyang Lake basin, China." Hydrological Processes **30**(14): 2562-2576.

---

## Author Comment (AC2) · 29 Apr 2020

The authors would like to thank the reviewers for their constructive comments that helped to improve the quality of the manuscript. Our point-by-point responses to the reviewers' valuable comments are listed below.

The comment: The manuscript is about using Budyko method and HBV model to investigate mean annual streamflow changes, due to climate variation and human influence, in the important Karkheh River Basin in western Iran. Although this manuscript is an interesting study but this study doesn't identify any major contributions in terms of process understanding or developing new methods. However as stated in the manuscript,

[Figure]

authors claim that their approach combining HBV and Budyko is novel and for the first time used in Iran. However, the knowledge gap/novelty and the importance of this work is still missing throughout the whole manuscript and it needs to be clearly stated.

Our response: Thank you for the detailed review and pinpointing shortcoming of the manuscript. Although, we admit that all the methods including the Budyko, HBV, DBEST, and satellite image processing used in this study are separately developed in previous studies, our manuscript introduces a novel combination of these methods such that a new, more robust framework of separating human vs climate variation effects on streamflow of large river basins in the data-scarce area is presented. As mentioned, this is the first time that Budyko analysis is implemented in the Iranian catchment. The case study is one of the most important catchments in the country and ironically there is the lack of studies investigating separate impacts of human activities and climate change in the area. In any case, the study offers a new approach for data scarce areas to quantify effects of climate change versus anthropogenic influence. This involves a technique to validate the Budyko method with remote sensing analyses. None of the previous studies mentioned by the reviewer has applied satellite remote sensing techniques to quantify land use changes over a long period of time for verification of the Budyko and HBV modelling results. We firmly believe that this is one of the major contributions of this study to the methodological approaches in the field of hydrology. Also, as a novel approach, we used the newly developed DBEST algorithm for detecting breakpoints. DBEST uses a (novel) segmentation algorithm for detecting and characterizing significant breakpoints, and has a general designation making it suitable for different time series data inputs. It has been applied successfully in several other studies and this paper shows its suitability for streamflow change detection. Not mentioning that the introduction of the DBEST method here is important because it is new in hydrological studies and specifically relevant in Budyko applications for a more reasonable, systematic selection of time periods to be compared.

The comment: The method section is too long with elaborated with details. In other

hand, the discussion (section is too short and general repeating the same message said earlier rather than putting the results from this study in a broader context of studies in similar regions and worldwide.

Our response: The methodology and introduction sections have been revised and present more precisely now. The discussion part was extended by adding more detailed discussion as suggested.

The comment: In the conclusion, authors claim that "The outcome of this study can be used to assist policymakers and water professionals in proposing a proper water management plan to prevent the further reduction of streamflow and groundwater storage". How the results of this study would help policy makers to prevent reduction of streamflow and groundwater storage? When the results show that we have a combined effect of both human (increased irrigated area and reduction of forests), and climate (decreasing annual precipitation) on streamflow reduction almost all over the basins.

Our response: Although it might be difficult to manage the impact of climate variation at the local scale, with a better understanding of the human activities' impact on water quantity, it is possible to introduce better management plans, such as improved agricultural management methods and urbanization control to limit the inverse impacts. These discussions are added to the revised manuscript.

The comment: Authors need to work properly with all figures for instance order of figures should be improved, authors refer to figure 2 and then figure 9 and then back to figure 3. Figure 11 can be removed from the discussion part and Figure 12 is not necessary.

Our response: Thank you for the suggestion. Your comment has been implemented in the revised version of the manuscript.

The comment: This a big assumption in this work that streamflow has not been influ-
enced by human activities before the breakpoint. Please clarify!

Our response: As mentioned earlier, the applied procedure is a common method which has been employed by other researchers such as (Hu, Liu et al. 2012; Sun, Tian et al. 2014; Chang, Zhang et al. 2016). However, your valuable point has now been discussed in the limitation sub-section on page14- line415, in the revised manuscript.

Additional References (added to the revised manuscript): Chang, J., H. Zhang, Y. Wang and Y. Zhu (2016). "Assessing the impact of climate variability and human activities on streamflow variation." Hydrology and Earth System Sciences 20(4): 1547-1560. Hu, S., C. Liu, H. Zheng, Z. Wang and J. Yu (2012). "Assessing the impacts of climate variability and human activities on streamflow in the water source area of Baiyangdian Lake." Journal of Geographical Sciences 22(5): 895-905. Sun, Y., F. Tian, L. Yang and H. Hu (2014). "Exploring the spatial variability of contributions from climate variation and change in catchment properties to streamflow decrease in a mesoscale basin by three different methods." Journal of Hydrology 508: 170-180. Wu, J., C. Miao, X. Zhang, T. Yang and Q. Duan (2017). "Detecting the quantitative hydrological response to changes in climate and human activities." Science of the Total Environment 586: 328-337. Zeng, S., J. Xia and H. Du (2014). "Separating the effects of climate change and human activities on runoff over different time scales in the Zhang River basin." Stochastic environmental research and risk assessment 28(2): 401-413. Zhang, Q., J. Liu, V. P. Singh, X. Gu and X. Chen (2016). "Evaluation of impacts of climate change and human activities on streamflow in the Poyang Lake basin, China." Hydrological Processes 30(14): 2562-2576.

Please also note the supplement to this comment: https://www.hydrol-earth-syst-sci-discuss.net/hess-2019-618/hess-2019-618-AC2-supplement.pdf

618, 2020.

---

## Author Comment (AC3) · 30 Apr 2020

Response to the reviewer's comments - hess-2019-618

The authors would like to thank the reviewers for their constructive comments that helped to improve the quality of the manuscript. Our point-by-point responses for the reviewers' valuable comments are listed below.

Reviewer 1: Comment: I have not identified any major contributions in this study whether in terms of process understanding or methods. I acknowledge that authors said that this is the first time the Budyko hypothesis is used in Iran, which is a positive that helps to improve the geographical bias in the hydroclimatology literature. That said, all the methods (breakpoint detection, Budyko hypothesis, HBV modeling) have been developed previously. Similar studies on fractional contribution of climate variation and human activities to annual runoff changes have been done before, such as (Liu et al., 2017) and (Yang & Yang, 2011), each offering a new contribution either in method or process understanding.

Answer: Thank you for the detailed review and pinpointing shortcoming of the manuscript. Although, we admit that all the methods including the Budyko, HBV, DBEST, and satellite image processing used in this study are separately developed in previous studies, our manuscript introduces a novel combination of these methods such that a new, more robust framework of separating human vs climate variation effects on streamflow of large river basins in the data-scarce area. As mentioned, this is the first time that Budyko analysis is implemented in an Iranian catchment. The case study is one of the most important catchments in the country and ironically there is the lack of studies investigating separate impacts of human activities and climate variation in the area. In any case, the study offers a new approach for data scarce areas to quantify effects of climate versus anthropogenic influence. This involves a technique to validate the Budyko method with remote sensing analyses. None of the previous studies mentioned by the reviewer has applied satellite remote sensing techniques to quantify land use changes over a long period of time for verification of the Budyko and HBV modelling results. We firmly believe that this is one of the major contributions of this study to the methodological approaches in the field of hydrology. Also, as a novel approach, we used the newly developed DBEST algorithm for detecting breakpoints. DBEST uses a (novel) segmentation algorithm for detecting and characterizing significant breakpoints, and has a general designation making it suitable for different time series data analysis. It has been applied successfully in several other filed of studies and this paper shows its suitability for streamflow change detection as well. Not mentioning that the introduction of the DBEST method here is important because it is new in hydrological studies and specifically relevant in Budyko applications for a more

reasonable, systematic selection of time periods to be compared.

Comment: Even compared to those studies, the present study is an over simplification when it comes to analyses such as estimation of potential evapotranspiration (PET) based on a just a correlation with temperature, inadequate model calibration with no account for parameter uncertainty, and inadequate sensitivity analysis of the Budyko hypothesis (section 3.5).

Answer: The sensitivity analysis provided in section 3.5 and Table 11 shows that the model is not very sensitive to changes in evapotranspiration and hence temperature. The primary objective here was to extend ET estimation covering the entire catchment. As the reviewer pointed out, we used a traditional, however, scientifically sound method for estimating ET at the basin level, having in mind that this was not the main focus of the study.

Comment: In addition to improving the analysis, the manuscript itself also requires major improvement: the paper is long with repetition of previous studies.

Answer: Thanks for the comment, we have done an intensive revision; we have shortened the manuscript and removed the repetitive sections highlighted in the revised version of the manuscript.

Comment: The present discussion (section 4) is just an overview of the study area, a proper discussion of the results of this study is required. In doing so, the results should be discussed within the a broader context of studies in similar regions in Iran or around the globe. The discussion section would also benefit from a new sub-section discussing the limitations of results and suggestions for future studies. There are several typos, hence a careful proof read.

Answer: The authors agree and the discussion section is improved in the revised version (page 12-14 – lines 345-445). Further, a sub-section was added to the discussion part, on page 14- lines 410-415, improving the analyses of the results. [Limitations of

the study: In order to discriminate the human activities and climate impacts on streamflow, in the hydrological approach, a couple of assumption was made. In this study, like similar studies (Zeng et al., 2014; Zhang et al., 2016; Wu et al., 2017), it was assumed that no human activities were involved in streamflow variation during the pre-change period. In other words, the human activities in pre-change period was considered negligible and the hydrological processes is natural. It was also assumed that climate variation and human activities are two independent variables, however as mentioned by Kim et al (2013) these two are dependent and it is not possible to correctly simulate LULC change without taking climate variation into account. (Kim et al, 2013, Dey and Mishra, 2017). Although the model uncertainty was not significant in this study, but caution must be applied in the interpretation of the findings. For instance, land use classification in the HBV model is rather simple with only three representatives. Catchment characteristic in Budyko method is assumed to be constant. As mentioned earlier, the catchment characteristic parameter is related to soil properties, slope and land use of the catchment, therefore it is subjected to change by changing LULC.]

Comment: One major conceptual point is that the terms “climate change” and “climate variation” are used interchangeably in the manuscript. Regardless of what term is better suited (maybe climatic changes?), it is important to clearly define in the manuscript (perhaps in the introduction) that in this study climate change/variation – which human activities are compared to – is a lump variable that encompasses both human-induced climate change and natural climatic variability. Therefore, the contribution of human activities are under-estimated. This is an important point to discuss in the discussion section (see my comment 1 for details). Use a consistent term in referring to climatic changes throughout the manuscript (including title). And clearly define that what you referred to as climate change/variation is in fact a lump variable that includes both human-induced climate change and natural climatic variabilities. Therefore, there is a contribution of human activities to climatic changes that you are not capturing in this study, and hence human activities, in general, are under-estimated in this framework. What you estimated as “human activities”, is in fact “regional human

activities/interventions" in terms of regional water resources projects which is distinct from emissions that are also due to human activities. Other factors such as the impact of dam construction on regional evaporation and precipitation is assumed to be zero, while a major dam as such would presumably have impacts on the regional climate.

Answer: Because the study period is not long enough to conclude climate change, the authors preferred to use the climate variability throughout the text. Definitely, human activities and climate change are tied together and not independent. However, to do the analysis, it is a common assumption to consider them as independent parameters (Zeng, Xia et al. 2014, Zhang, Liu et al. 2016, Wu, Miao et al. 2017). In this study, as you mentioned, climate variability is a combination of variability of climate, as well as global-scale human-induced effects due to worldwide greenhouse gasses increase. Human activities, on the other hand, is regional impacts of human. Dam construction and agricultural activities, etc. are intimately embedded in human impact. Many of these relationships, now have been better described and discussed in the revised manuscript.

Comment: Human activities are also under-estimated within the Budyko model by definition, as the parameter n in the Budyko equation is assumed to be a constant. That is, interannual changes in catchment characteristics are assumed to be zero (Yang & Yang, 2011).

Answer: To calculate the parameter n similar studies (e.g., Hu, Liu et al. (2012)) have used the same approach. As discussed by Liang et al (2015)"Given P, Q, and E0, we estimated parameter n by modeling E (Eqn.1) and at the same time minimizing the difference between modeled E and E estimated from the long-term catchment water balance while neglecting changes in soil water storage (Eqn.2)". E/P= (E0/P)/ãĂŰ[1+(Eo/p)ˆn]ãĂŮˆ(1/n) Eqn.1 E = P –Q Eqn.2

However, we have now discussed this limitation (the constant n) on page 15- line 425.

Comment: That means the pre-change period could have been impacted by regional

human activities that are hard to capture and compare with the post-change period within the present method. a. On this topic, there been several recent studies on Lake Urmia, not far from KRB (Alizade Govarchin Ghale et al., 2018; Alizade Govarchin Ghale et al., 2019; Chaudhari et al., 2018; Fazel et al., 2017; Khazaei et al., 2019). Fazel et al. (2017) reported an interesting point, relevant to your study, that “flow regime in river headwaters appeared to be dominated by natural forces, . . . the reduction in river flow magnitude increased from headwaters to downstream reaches for all rivers”. This is a major point to bear in mind when discussing your results, as you have not differentiated between headwater and downstream changes in flow. 2.

Answer: Definitely upper flow and downstream play important roles in a catchment’s flow regime and we have considered them too, in this work. The basin was divided into five sub-basins and the upper basin streamflow was taken into account while doing analysis for the lower sub-basin.

Comment: The introduction requires further refinement to be sharp and concise. 3. There is no shortage of method for estimating potential evapotranspiration, including plenty of temperature-based methods (McMahon et al., 2013). A simple correlation is an oversimplification. The T-PET is not a linear relationship. Check out the modified Hargreaves method Droogers and Allen (2002) or the temperature-based method in Oudin et al. (2005). 4.

Answer: We have improved the introduction. As mentioned, our results are not that sensitive to changes in evaporation. Moreover, there are several evapotranspiration stations unevenly distributed across the catchment, thus we only extended the ET estimation using the described method.

Comment: I’d take issue with your hydrological modelling approach on two main grounds. (1) Parameter uncertainty is not accounted for. It is well-established that hydrological models require some type of parameter uncertainty analysis, i.e. models should be used as ensembles.

Answer: We have now changed the previous HBV modelling results and calibrated the most recent version of HBV model (HBV-light) which provides a tool to integrate the "uncertainty analysis". Using the embedded Genetic Algorithm in the HBV-light model, we were able to perform an uncertainty analysis using the 10 best parameter sets. The results of the investigated uncertainty is added in the revised version of the manuscript (subsection on page 12).

Comment: In addition, it is not clear whether you have calibrated the model on daily scale or monthly. Calibrating the model on monthly scale would introduce additional uncertainties. It is easier to achieve a higher model performance in monthly calibrations, as daily variations would be ignored. This is particularly important when models are transferred to changing conditions, as in this study. A model that cannot reproduce daily variations in the calibration period, would do even worse during an evaluation period with considerable changed conditions.

Answer: The HBV model simulation including calibration process and all inputs are on a daily basis. In the figures, they are shown on a monthly time-scale for the sake of presentation. We have now discussed this for clarification on page 9 line 279: [It is noted that all the evaluation indices in Table 7 (The calibration performance indices for the sub-basins) are calculated based on a daily data, while Figure 7 shows the annual-based values for the sake of presentation.]

Comment: (2) The objective functions used to evaluate the model performance are inadequate. R2, which is essentially NSE (Nash Sutcliffe efficiency), has been demonstrated to be an inadequate metric for evaluating model performance (Gupta et al., 2009; Murphy, 1988). KGE (Kling Gupta Efficiency, updated version by Kling et al. (2012)) has been shown to be a better alternative to NSE, which already includes a bias term. So, if KGE is used, the second metric you used (Equation 3) would be redundant and you can remove it from model calibration/evaluation. So, I'd suggest to redo the hydrological modeling to account for parameter uncertainty using the KGE metric – and on a daily basis.

Answer: Thank you for the suggestion. We recalibrated the hydrological model using the new version of HBV model and added the mentioned criteria (KGE) to Table 8, based on your suggestion.

Comment: a. Details about HBV model particularly the name of the parameters are unnecessary. Simply remove them and refer to original literature of "the version" of the model you used in this study (there are different versions of HBV). Other parts such as Table 7 (range of calibrated parameters) should be in the supplements as they are not contributing to the main story of the paper.

Answer: Thanks for the comment. We have now revised the manuscript accordingly. The introduction is enhanced and is more specific now. Unnecessary and wordy parts have been removed and the table has been moved to Appendix section on page 33.

Comment: Lines 356-360 "The HBV model overestimated the streamflow for the post-change period, which suggests that there existed factors other than climate variations affecting the streamflow of the study basins. These factors are believed to be related to human activities". You are oversimplifying the problem here. The issue that hydrological models perform poorly under changing conditions is a widely recognized issue and in fact one of the unsolved question of hydrological sciences (Question 19 in Blöschl et al., 2019). That said, you cannot simply conclude that those changes are due to human activities. The underlying reason is a combination of model structure adequacy to represent the evolution of catchment processes, model parameterization, climatic changes (whether natural variability or human-induced) that cannot be represented well by model structure and parameters, and changes due to human activities. I agree that human activities has a role in this, but you cannot simply "believe" that it is all due to human activities.

Answer: Definitely, different scholars implement different methods based on their perceptions and expertise. However, as mentioned by Dey et al. (2017), the applied procedure is a common method, which has been employed numerous studies including Hu, Liu et al. (2012); Sun, Tian et al. (2014); Chang, Zhang et al. (2016). However, your valuable point has now been discussed in the limitation section in the revised manuscript.

Comment: 5. Section 2.5 on Budyko method is too long. All the equations have been derived previously. You can simply explain the method conceptually, direct readers to the literature for more details, and present only the equations you used. a. Line 241, it is not “believed” but “assumed” to be. . .

Answer: Thanks for the comment. We revised the manuscript accordingly. ‘Believed’ was also changed to ‘assumed’.

Comment: 6. Section 2.7, lines 288-289, “In order to adequately simulate a hydrological response at the basin level, accurate data such as climate variables (precipitation, ET, etc.) and catchment physical characteristics (topography, land coverage, vegetation, etc.) are vital.” what about the adequacy of the model structure, both the Budyko model and HBV, in representing catchment processes and change? a. Lines 293-294, it does not make sense and does not follow the previous paragraph.

Answer: Thanks for the comment. We reviewed and changed these lines and highlighted them in the text, uncertainty analysis subsection.

Comment: The possible sources of errors (line 300) are the break change method, Budyko model and parameterization, HBV model structure and parameterization, and data quality and length. Each of these factors, individually or combined, may change the results to some degree. Among these factors, you did not look into the dependency of your results upon the choice of breakpoint method. For instance Liu et al. (2017) used 8 different methods to account for this.

Answer: Definitely, each hydrological model has its own advantages and limitations and Budyko and HBV models are no exceptions. Liu et al. (2017) studied several catchments across china, using three approaches of AMOC, Ordered Clustering Test,

and Pettitt test for the detection of change points. The focus of studies such as Liu et al. (2017) is to analyse, in detail, the driving factors and their exact shares affecting streamflow changes. While, in this study we primarily focus on the impact of human activities vs. climate variation. For future studies, if there is enough data available, the details of human activities can be specifically studied. The breaking point occurs when the trend of the streamflow changes. In this study, the breakpoints were detected by a newly developed DBEST method and was cross validated with Man-Kendall test.

Comment: What you discussed in section 3.5 is actually a sensitivity analysis of the Budyko model, for which you only reported 10% change in parameter n. What if all the inputs and parameters of Budyko are subjected to 5, 10 and 15% change? How the change in Budyko inputs propagate to the final answer? What about the change in model structure, e.g. if another model was used instead of HBV?

Answer: The suggestion for future work and discussion has now been added to the discussion part. Implementing all mentioned methods are beyond the scope of this research.

Comment: The discussion of land use changes (section 3.4) can be improved. It takes one page, two figures and two tables to just say that irrigation area has increased significantly while forested area decreased. It is also good to briefly mention that there are three snapshots of an early year, changepoint, and final year to help readers follow the story line.

Answer: Thank you for your suggestion. It has now been added to the revised manuscript line 312: [These three years represent the KRB land use condition for three phases of before breakpoint, breakpoint, and after breakpoint, respectively.]

Comment: The figures are not properly ordered. After presenting figure 2 on page 5, authors jumped to figure 9 on page 10, and then back to figure 3 (page 10). Figure 11 is not relevant to the discussion session of the paper, it is relevant to introduction or study area. Figure 12 does not offer anything new, Yang and Yang (2011) already

presented and discussed this. a. Figure 3, present all the 5 sub-basins in one figure. b. Figure 6 is not easy to read. c. Figure 7 and 8 could be combined, as two subplots.

Answer: Thanks for the precise comments. We amended the text and figures accordingly.

Comment: 9. Line 469, “The Budyko method showed to be a reliable . . . method to analyze streamflow changes during the study period”. You have not offered any analysis nor discussion on the reliability of the Budyko method. There are several typos throughout the text. A carful proof read is required. Here are a few minor corrections and typos (just as examples): a. The name of “Seimareh” river has two different spellings on the Figure and in the text. b. The symbol for changes due to human activities is not consistent: ∆ðİŚĎðİŘżðİŘť (e.g. line 243) and ∆ðİŚĎðİŘż (e.g. equation 11) – fix throughout. Also fix the symbol for ET0 and E0 in equation 13. And the denominator in the second term of the equation 15. c. Avoid making up abbreviations in the middle of the manuscript such as Eqn on line 202. Be consistent either use abbreviation or the full word throughout. Check the journal guideline for this. Same comment applies to Fig. and Figure.

Answer: We appreciate your careful reading of the manuscript. These comments are now implemented and highlighted in the revised version of the manuscript.

Comment: a. Within the current visualization of Figure 1, it is very hard to see the stream gauges in most sub-basins. Since you didn’t mark each discharge station with its name on the map, it is hard for readers to tell where Payepol station is: is it the blue square inside the Upper Karkheh sub-basin, or is it the one outside of it (the final outlet the entire catchment)? Also show the location of the Karkheh dam on the figure.

Answer: Thanks for pinpointing the issues. An improved map has now replaced the previous one.

Additional References (added to the revised manuscript): Chang, J., H. Zhang, Y.

Wang and Y. Zhu (2016). "Assessing the impact of climate variability and human activities on streamflow variation." Hydrology and Earth System Sciences 20(4): 1547-1560. Hu, S., C. Liu, H. Zheng, Z. Wang and J. Yu (2012). "Assessing the impacts of climate variability and human activities on streamflow in the water source area of Baiyangdian Lake." Journal of Geographical Sciences 22(5): 895-905. Kim, J., J. Choi, C. Choi and S. Park.: Impacts of changes in climate and land use/land cover under IPCC RCP scenarios on streamflow in the Hoeya River Basin, Korea, Science of the Total Environment 452: 181-195, 2013. Liang, W., Bai, D., Wang, F., Fu, B., Yan, J., Wang, S., and Feng, M.: Quantifying the impacts of climate change and ecological restoration on streamflow changes based on a Budyko hydrological model in China's Loess Plateau. Water Resour. Res., 5, 6500–6519, https://doi.org/10.1002/2014WR01658, 2015. Sun, Y., F. Tian, L. Yang and H. Hu (2014). "Exploring the spatial variability of contributions from climate variation and change in catchment properties to streamflow decrease in a mesoscale basin by three different methods." Journal of Hydrology 508: 170-180. Wu, J., C. Miao, X. Zhang, T. Yang and Q. Duan (2017). "Detecting the quantitative hydrological response to changes in climate and human activities." Science of the Total Environment 586: 328-337. Zeng, S., J. Xia and H. Du (2014). "Separating the effects of climate change and human activities on runoff over different time scales in the Zhang River basin." Stochastic environmental research and risk assessment 28(2): 401-413. Zhang, Q., J. Liu, V. P. Singh, X. Gu and X. Chen (2016). "Evaluation of impacts of climate change and human activities on streamflow in the Poyang Lake basin, China." Hydrological Processes 30(14): 2562-2576.

Please also note the supplement to this comment:
https://www.hydrol-earth-syst-sci-discuss.net/hess-2019-618/hess-2019-618-AC3-supplement.pdf

---

## Author Comment (AC4) · 30 Apr 2020

Response to the reviewer's comments - hess-2019-618

The authors would like to thank the reviewers for their constructive comments that helped to improve the quality of the manuscript. Our point-by-point responses for the reviewers' valuable comments are listed below.

Reviewer 2: Comment: The manuscript is about using Budyko method and HBV model to investigate mean annual streamflow changes, due to climate variation and human influence, in the important Karkheh River Basin in western Iran. Although this

manuscript is an interesting study but this study doesn't identify any major contributions in terms of process understanding or developing new methods. However as stated in the manuscript, authors claim that their approach combining HBV and Budyko is novel and for the first time used in Iran. However, the knowledge gap/novelty and the importance of this work is still missing throughout the whole manuscript and it needs to be clearly stated.

Answer: Thank you for the detailed review and pinpointing shortcoming of the manuscript. Although, we admit that all the methods including the Budyko, HBV, DBEST, and satellite image processing used in this study are separately developed in previous studies, our manuscript introduces a novel combination of these methods such that a new, more robust framework of separating human vs climate variation effects on streamflow of large river basins in the data-scarce area. As mentioned, this is the first time that Budyko analysis is implemented in an Iranian catchment. The case study is one of the most important catchments in the country and ironically there is the lack of studies investigating separate impacts of human activities and climate variation in the area. In any case, the study offers a new approach for data scarce areas to quantify effects of climate versus anthropogenic influence. This involves a technique to validate the Budyko method with remote sensing analyses. None of the previous studies mentioned by the reviewer has applied satellite remote sensing techniques to quantify land use changes over a long period of time for verification of the Budyko and HBV modelling results. We firmly believe that this is one of the major contributions of this study to the methodological approaches in the field of hydrology. Also, as a novel approach, we used the newly developed DBEST algorithm for detecting breakpoints. DBEST uses a (novel) segmentation algorithm for detecting and characterizing significant breakpoints, and has a general designation making it suitable for different time series data analysis. It has been applied successfully in several other filed of studies and this paper shows its suitability for streamflow change detection as well. Not mentioning that the introduction of the DBEST method here is important because it is new in hydrological studies and specifically relevant in Budyko applications for a more
reasonable, systematic selection of time periods to be compared.

Comment: The method section is too long with elaborated with details. In other hand, the discussion (section is too short and general repeating the same message said earlier rather than putting the results from this study in a broader context of studies in similar regions and worldwide.

Answer: The methodology and introduction sections have been revised and improved. The discussion part was extended by adding more detailed discussion as suggested – page 13-14- line 370 -450, as follow:

[Uncertainty analysis: The HBV modelling results suggested that the proposed parameters provided by Genetic Algorithm (GA) could model the catchments with reasonably well accuracy. However, the model may suffer from some uncertainties due to input observed data or model structure. In order to adequately simulate a hydrological response at the basin level, accurate data such as climate variables (precipitation, ET, etc.) and catchment physical characteristics (topography, land coverage, vegetation, etc.) are vital. In climate variation related studies, in which the study period is on the scale of decades, it is difficult to obtain uniformly distributed and accurate data sets (Kapangaziwiri et al. 2009). In the Karkheh catchment, specifically, part of the uncertainty may arise from observed rainfall, potential evapotranspiration and streamflow data. The weather gauges and stations are not uniformly distributed in the entire catchment. Moreover, elevation and topography of the catchment can introduce bias to the observation time series, which can subsequently affect runoff simulation and modelling (Rientjes et al. 2013). Another sources of uncertainty may arise from non-uniqueness of the model parameters, which means that different combination of parameters may result in the same streamflow prediction (Beven. 2001; Masih et al. 2010). In this study, the non-uniqueness of the model parameters was also investigated. To do so, the 10 best sets of calibration parameters produced by GA were selected for each sub-basin to estimate the impact of human activities and climate variation on streamflow variation. The results appear to be reliable if the streamflow prediction remains consistent,

despite the change in parameters (Masih et al. 2010). Table 9 shows the streamflow simulation is consistent despite the change in parameter sets.]

[Limitations of the study: In order to discriminate the human activities and climate impacts on streamflow, in the hydrological approach, a couple of assumption was made. In this study, like similar studies (Zeng et al., 2014; Zhang et al., 2016; Wu et al., 2017), it was assumed that no human activities were involved in streamflow variation during the pre-change period. In other words, the human activities in pre-change period was considered negligible and the hydrological processes is natural. It was also assumed that climate variation and human activities are two independent variables, however as mentioned by Kim et al (2013) these two are dependent and it is not possible to correctly simulate LULC change without taking climate variation into account. (Kim et al, 2013, Dey and Mishra, 2017). Although the model uncertainty was not significant in this study, but caution must be applied in the interpretation of the findings. For instance, land use classification in the HBV model is rather simple with only three representatives. Catchment characteristic in Budyko method is assumed to be constant. As mentioned earlier, the catchment characteristic parameter is related to soil properties, slope and land use of the catchment, therefore it is subjected to change by changing LULC.]

Comment: In the conclusion, authors claim that "The outcome of this study can be used to assist policymakers and water professionals in proposing a proper water management plan to prevent the further reduction of streamflow and groundwater storage". How the results of this study would help policy makers to prevent reduction of streamflow and groundwater storage? When the results show that we have a combined effect of both human (increased irrigated area and reduction of forests), and climate (decreasing annual precipitation) on streamflow reduction almost all over the basins.

Answer: Although it might be difficult to manage the impact of climate variation at the local scale, with a better understanding of the human activities' impact on water quantity, it is possible to introduce better management plans, such as improved agricultural

management methods and urbanization control to limit the inverse impacts. These discussions are added to the revised manuscript.

Comment: Authors need to work properly with all figures for instance order of figures should be improved, authors refer to figure 2 and then figure 9 and then back to figure 3. Figure 11 can be removed from the discussion part and Figure 12 is not necessary.

Answer: Thank you for the suggestion. Your comment has been implemented in the revised version of the manuscript. We fixed the order of the figures in the revised manuscript. We also removed figure 12 and relocated figure 11 to the study area subsection – page 3 – line 90

Comment: This a big assumption in this work that streamflow has not been influenced by human activities before the breakpoint. Please clarify!

Answer: As mentioned by Dey et al. (2017), the applied procedure is a common method, which has been employed numerous studies including Hu, Liu et al. (2012); Sun, Tian et al. (2014); Chang, Zhang et al. (2016). However, your valuable point has now been discussed in the limitation section in the revised manuscript.

Additional References (added to the revised manuscript): Chang, J., H. Zhang, Y. Wang and Y. Zhu (2016). "Assessing the impact of climate variability and human activities on streamflow variation." Hydrology and Earth System Sciences 20(4): 1547-1560.

Hu, S., C. Liu, H. Zheng, Z. Wang and J. Yu (2012). "Assessing the impacts of climate variability and human activities on streamflow in the water source area of Baiyangdian Lake." Journal of Geographical Sciences 22(5): 895-905. Kim, J., J. Choi, C. Choi and S. Park.: Impacts of changes in climate and land use/land cover under IPCC RCP scenarios on streamflow in the Hoeya River Basin, Korea, Science of the Total Environment 452: 181-195, 2013.

Liang, W., Bai, D., Wang, F., Fu, B., Yan, J., Wang, S., and Feng, M.: Quantifying the impacts of climate change and ecological restoration on streamflow changes based

on a Budyko hydrological model in China's Loess Plateau. Water Resour. Res., 5, 6500–6519, https://doi.org/10.1002/2014WR01658, 2015.

Sun, Y., F. Tian, L. Yang and H. Hu (2014). "Exploring the spatial variability of contributions from climate variation and change in catchment properties to streamflow decrease in a mesoscale basin by three different methods." Journal of Hydrology 508: 170-180.

Wu, J., C. Miao, X. Zhang, T. Yang and Q. Duan (2017). "Detecting the quantitative hydrological response to changes in climate and human activities." Science of the Total Environment 586: 328-337.

Zeng, S., J. Xia and H. Du (2014). "Separating the effects of climate change and human activities on runoff over different time scales in the Zhang River basin." Stochastic environmental research and risk assessment 28(2): 401-413.

Zhang, Q., J. Liu, V. P. Singh, X. Gu and X. Chen (2016). "Evaluation of impacts of climate change and human activities on streamflow in the Poyang Lake basin, China." Hydrological Processes 30(14): 2562-2576.

Please also note the supplement to this comment:
https://www.hydrol-earth-syst-sci-discuss.net/hess-2019-618/hess-2019-618-AC4-supplement.pdf